# Entanglement Dynamics of Random GUE Hamiltonians

**Daniel Chernowitz[1][*] and Vladimir Gritsev[1,2]**

**1** Institute for Theoretical Physics, Universiteit van Amsterdam,
Science Park 904, Postbus 94485, 1098 XH Amsterdam, The Netherlands
**2** Russian Quantum Center, Skolkovo, Russia

[*] d.m.chernowitz@uva.nl

## Abstract

In this work, we consider a model of a subsystem interacting with a reservoir and study dynamics of entanglement assuming that the overall time-evolution is governed by non-integrable Hamiltonians. We also compare to an ensemble of Integrable Hamiltonians. To do this, we make use of unitary invariant ensembles of random matrices with either Wigner-Dyson or Poissonian distributions of energy. Using the theory of Weingarten functions, we derive universal average time evolution of the reduced density matrix and the purity and compare these results with several physical Hamiltonians: randomized versions of the transverse field Ising and XXZ models, Spin Glass and, Central Spin and SYK model. The theory excels at describing the latter two. Along the way, we find general expressions for exponential $n$-point correlation functions in the gas of GUE eigenvalues.

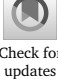

# 1   Introduction

The question of emergence of thermal behavior in isolated quantum systems has attracted considerable attention since the birth of quantum mechanics [1]. Recently intensive interest from the community has been reawakened. In part it is motivated by the availability of new experimental systems (see e.g. [2–11] for recent cold atomic experiments) that enable us to probe thermalization or its absence through precise control of microscopic conditions. Studies of these experiments also deepened our theoretical understanding of the universal laws governing dynamics of generic many-body systems and constraints in the contrasting case of integrability.

There are several roads leading to thermalization in quantum systems: one is based on the eigenstate thermalization hypothesis (ETH) [12–16] and is supported by a body of numerical evidence. It asserts that sufficiently complex quantum system (in particular ones that are chaotic in the classical limit) have eigenstates that are essentially indistinguishable from thermal states with the same average energy. A similar claim holds for local operator averages of almost all observables. In other words, a global pure quantum state is apparently indistinguishable from a mixed, globally-entropic thermal ensemble. In this respect it is interesting to understand *dynamics* of the process of thermalization in generic closed quantum systems.

A different path to accessing *universal* features of thermalization is to assume that dynamics are governed by a random Hamiltonian which smears out all microscopic detail. Assuming some overall symmetries it can be postulated further that the randomized Hamiltonian is described by one of the traditional random matrix ensembles (GUE, GOE, or GSE). This type of setup has been implemented in several recent works [17–27], and most notably, [28]. In all deference, this last paper predates the current work and shares the same perspective. In it,

a number of our results were already independently established, along with some interesting supplementary ones.

Among all of the recent works, quantum states converge dynamically to agree with the predictions of a thermal ensemble, and the universal quantities accompanied by this convergence have been established. Most of the above papers focus however on the asymptotic limit of very large or even infinite Hilbert spaces. The perspective of the present work is on explicit dependence on the finite system size.

While one can argue that the Hamiltonians drawn from one of the random matrix ensembles correspond to somewhat unphysical situation, we believe that our results are applicable to a large class of physically relevant models where long-range interactions dominate. For example, this is realized in central-spin type models whose Hamiltonians $H_{CS} = \sum_k J_k \vec{S}_0 \vec{I}_k$ describe interaction between a "central spin" $\vec{S}_0$ and the nuclear spins $\vec{I}_k$ with arbitrarily distributed couplings $J_k$. This Hamiltonian, while integrable for arbitrary $J_k$'s, is used to model quantum dots and NV centers [29, 30] reduced BCS models and Dicke-type systems [31]. These integrable models represent a subclass of more elaborate non-integrable counterparts which can be used to describe more realistic setups [32]. The latter then are used to describe, for example, dipole-dipole interacting spin models [33] used to describe e.g. ions in a trap or nitrogen vacancy centers [34, 35] which were instrumental in observing time crystals. Recently, the random Hamiltonian setup or random quantum circuits have been used for studying universal features of the out-of-time correlation function [23, 26, 36, 37], entanglement features [27], unitary design [25] and spectral decoupling [23], as well as entanglement tsunami [38], and measurement induced phase transitions [39].

Having in mind all these recent developments, here we consider a model of a subsystem interacting with a reservoir. We study dynamics of information transfer in this system assuming that the overall system is described by non-integrable or integrable Hamiltonians with uniformly random spectral basis, which are modeled by RMT from either Wigner-Dyson or Poissonian distributions.

The paper is organized as follows. In the next section, the technical setup of the subsystem/reservoir is explained, as well as the mathematical notation. The random matrix models provide a distribution of systems, over which we must average. This averaging is done by integrating over the degrees of freedom of the matrices, and similar to a polar coordinate system on the plane, this integral is split into two parts. The *angular* part is treated in section 3, and the *radial* part in section 4. Along the way, we find quite general procedures to average functions of reduced density matrices, and also for vertex operator-esque correlators in the field theory of random matrix eigenvalues. Then in section 5, the results of these averages are discussed, their properties highlighted, and notably they are compared to numerics by sampling the studied matrix ensembles, and to ensembles of existing famous models from (one-dimensional) condensed matter theory. In the final section, some outlook and discussion is offered.

# 2 Setup

We are interested in using Random Matrix Theory (RMT), specifically the techniques of unitary integrals and correlation functions, to gain insights in the statistics of bipartite discrete quantum systems, induced by a Gaussian ensemble of Hamiltonians.

## 2.1 Bipartite Systems

The setup is as follows. We consider a discrete finite full quantum system $A + B$ consisting of a subsystem $A$, and a bath $B$, with a tensor product Hilbert space $\mathcal{H} = \mathcal{H}_A \otimes \mathcal{H}_B$ of dimension

$d_A \times d_B = d$. On the full space, a constant Hamiltonian, a $d \times d$ Hermitian operator or matrix, governs time evolution. Notably, this Hamiltonian is thought to be a general random interaction on $\mathcal{H}$, and has no knowledge of locality or the division into $A$ and $B$. The partition simply arises because the experimenter has access to $A$ via local observables $O$: Hermitian operators that act as the identity on $B$: $O = O_A \otimes \mathbb{1}_B$. $B$ may be thought of as much larger than $A$, but this is not crucial. We will repeatedly make use of the mapping of a Hermitian matrix to its eigenvalues and eigenvectors

$$(V, E) \leftrightarrow H = V \Lambda V^\dagger, \quad \Lambda := \begin{pmatrix} E_1 & & & \\ & E_2 & & \\ & & \ddots & \\ & & & E_d \, , \end{pmatrix}, \tag{1}$$

found by diagonalizing $H$. In this context, this is known as the *radial-angular-decomposition* [40]. Here $E = \{E_j\}$ is a vector of real eigenvalues (energies) and $V \in \mathcal{U}(d)$ is a unitary matrix encoding the spectral basis[1].

One assumption made in this work is that at time $t = 0$ the subsystem and bath are brought into contact with each other, resulting in a full system product state $|1\rangle = |1_A\rangle \otimes |1_B\rangle = |1_A; 1_B\rangle$: the constituent systems are initially pure[2]. The choices of $|1_A\rangle$ and $|1_B\rangle$ are arbitrary, but our notation will use a basis in which $|1\rangle$ is the first basis vector[3].

At later times, by the Schrödinger equation, the state is

$$|t\rangle := e^{-iHt} |1\rangle = \sum_{n=1}^{\infty} \frac{(it)^n}{n!} \left( V \Lambda V^\dagger \right)^n |1\rangle = V e^{-i\Lambda t} V^\dagger |1\rangle. \tag{2}$$

$V$ also diagonalizes the time-evolution operator, so $|t\rangle$ is polynomial in $V, V^\dagger$. In component notation, with completed basis $\{|k\rangle\}$ of $\mathcal{H}$, it looks like

$$|t\rangle = \sum_{j,k=1}^{d} |k\rangle V_{kj} e^{-iE_j t} V_{j1}^\dagger. \tag{3}$$

From here, we can obtain the mixed state description of the subsystem $A$ by tracing out the bath,

$$\rho_A(t) = \text{Tr}_B \left( |t\rangle \langle t| \right). \tag{4}$$

Then, implicitly $\rho_A$ is thus also a function of $V$ and $E$. In the basis $\{|h_A\rangle\}$ of $\mathcal{H}_A$, the elements of this matrix are given by

$$\langle g_A | \rho_A(t) | h_A \rangle = \sum_{h_B=1}^{d_B} \sum_{j,k=1}^{d} V_{(g_A, h_B)j} e^{-iE_j t} V_{j1}^\dagger V_{1k} e^{iE_k t} V_{k(h_A, h_B)}^\dagger. \tag{5}$$

The expression is visualized in a diagram in figure 1. The summation over $h_B$ results from tracing out the bath. This is done by expanding the outer indices of $\rho = |t\rangle \langle t|$ into a double-index, not unlike the digits of a decimal number: a bath state tensored with a subsystem state $|h\rangle \langle g| = |h_A; h_B\rangle \langle g_A; g_B|$, and contracting with $\delta_{h_B, g_B}$. Others, namely $j, k$, are internal full indices from the matrix multiplication of the diagonal decomposition, ranging over the

---

[1]This mapping is not exactly bijective, $H$ is overcounted because right-multiplying $V$ by a permutation matrix or an element of $\mathcal{U}(1)^{\otimes d}$ results in the same Hamiltonian. However, in the results below this will be compensated by normalization. This symmetry is larger for degenerate systems but these are measure zero in the full space, so they do not affect the statistics either.

[2]Or we could imagine performing a selective measurement on $A$ at $t = 0$ to the same effect.

[3]$|1\rangle$ is not an eigenstate of $H$.

$$\text{Tr}_B : h_B$$

Figure 1: Diagram of expression (5). Lines indicate matrix multiplication, with summed index below. The outer indices are split into a subsystem and bath double-index, the trace sums over the latter.

full dimensionality of $\mathcal{H}$. In what follows, we will use the Einstein summation convention on repeated indices[4]. In doing so, we remember that indices with a subscript $A$ (e.g. $g_A$) are summed from 1 to $d_A$, subscript $B$ summed to $d_B$ and unsubscripted to $d$.

Local observables $O_A$ on $A$ take the form of Hermitian operators on $\mathcal{H}_A$, and an expectation value can be found: $\bar{o} = \text{Tr}_A(O_A \rho_A)$. This is the main utility of $\rho_A$.

Another important quantity in this work is the *purity* [41]:

$$\gamma := \text{Tr}_A \rho_A^2. \tag{6}$$

It contains information about the entanglement between $A$ and $B$: The lower the purity, the higher the entanglement. A product state like $|1\rangle$ gives $\rho_A = |1_A\rangle \langle 1_A|$, for which $\gamma = 1$. On the other hand, an entangled state cannot be written in this product form and will have $\gamma < 1$. The lowest purity is $1/d_A$, corresponding to the maximally mixed $\rho_A = \mathbb{1}_A/d_A$. Though less studied than the Von Neumann entropy, it is computationally favorable because it does not involve diagonalization of $\rho_A$ or an infinite power series (the logarithm).

## 2.2 GUE Distribution and Measure

In random matrix theory, one treats matrices as random variables. They can be integrated, and thus averaged over, if one takes care constructing the measure and probability density function. We will use the Gaussian Unitary Ensemble (GUE) in the following. The weight on the eigenvalues will be multivariate Gaussian,

$$\mathcal{P}(E) = \mathcal{C} \exp\left(-\frac{\lambda}{2} \sum_{l=1}^{d} E_l^2\right), \tag{7}$$

with $\mathcal{C}$ a normalization constant and $\lambda$ set to 1 in this work. The measure is given by $\Delta^2(E)\mathcal{D}E dV$.

$$\Delta(E) := \det_{1 \leq j,k \leq d}(E_j^{k-1}) = \prod_{1 \leq j < k \leq d} (E_j - E_k) \tag{8}$$

is the Vandermonde determinant, the Jacobian of the transformation to angular-radial coordinates. $\mathcal{D}E$ is the product of Lebesgue measures on the eigenvalues, and $dV$ is the Haar measure on the Unitary group $\mathcal{U}(d)$. Some explanatory background information, as well as the theory needed to integrate out these measures, is collected in appendix A.

## 3 Angular Integral: Unitary Average over Eigenstates

Now we make our first steps towards non-standard calculations. In the setup, we have cited a probability density function (PDF) and measure $\mathcal{P}(E)\Delta^2(E)DV\mathcal{D}E$ on $\mathcal{M}$, the space of possible

---

[4]Also triple indices are summed over, this is done rather than include the Kronecker delta and an additional label.

Hamiltonians. The next step will be to choose a suitable integrand to average over this PDF. The choice we make, is the reduced density matrix $\rho_A(t)$. In this section, we will perform the equivalent of the angular integral in polar coordinates: we will average over the compact unitary group $\mathcal{U}(d)$. A note on notation: when an average of some $f$ has been carried out over $dV$ or $\mathcal{D}E$ we will write $\langle f \rangle$. If both have been done, we will use double brackets: $\langle\langle f \rangle\rangle$. The authors would like to be clear that although the results of this section were achieved independently, most if not all were already established some years earlier in [28].

## 3.1 Reduced Density Matrix

Returning to expression (5) and taking time $t$ as a fixed parameter, we first wish to average (integrate) $\rho_A(t)$ over the eigenbasis $V \in \mathcal{U}(d)$. We will average element-wise, so the result is again a $(d_A \times d_A)$ matrix. The salient observation is that each element of this matrix $\rho_A$ is polynomial in the elements of $V$ and $V^\dagger$. In fact it is second order in both. As we vary $V$, we are interested in individual terms of expression (5), which are $\int_{\mathcal{U}(d)} dV V_{(g_A,h_B)j} V_{1k} V_{j1}^\dagger V_{k(h_A,h_B)}^\dagger$. This is handled using the theory of Weingarten functions, see appendix A for details.

In the language of equation (58), the sets of indices are:

$$
\begin{aligned}
I &\equiv (i_1, i_2) = \big((g_A, h_B), 1\big); \quad I' \equiv (i_1', i_2') = \big(1, (h_A, h_B)\big) \\
J &\equiv (j_1, j_2) = (j, k); \qquad\qquad J' \equiv (j_1', j_2') = (j, k).
\end{aligned}
\tag{9}
$$

The inner product is linear, so we may write

$$
\langle g_A | \langle \rho_A \rangle | h_A \rangle := \int_{\mathcal{U}(d)} dV \langle g_A | \rho_A | h_A \rangle = \sum_{\sigma, \tau \in S_2} \delta_{I, \sigma(I')} \delta_{J, \tau(J')} \mathrm{Wg}(d, \sigma\tau^{-1}) e^{i(E_k - E_j)t}.
\tag{10}
$$

Here, $S_2$ is the symmetric group on two symbols, and $\delta_{I, \sigma(I')} := \prod_l \delta_{I_l, I_{\sigma_l}'}$. Pulling this all together, a consistent pattern emerges: the $I, I'$-terms decouple from the $J, J'$-terms into a product structure: they do not depend on each other, neither in the contributions due to equation (58), nor in the multiplicative factors that appear from the Schrödinger equation. This is a feature that appears repeatedly in our unitary average calculations, so it warrants some notation

$$
\langle \rho_A \rangle = \sum_{\sigma, \tau \in S_q} R_\sigma Q_\tau \mathrm{Wg}(d, \sigma\tau^{-1}).
\tag{11}
$$

The objects $R_\sigma$ and $Q_\tau$ can be seen as vectors of length $q!$, with elements indexed by the permutations of $S_q$. In the present case, $\rho_A$ is quadratic in $V$ and $V^\dagger$, so $q = 2$, and the former are[5]:

$$
R_\sigma := |g_A\rangle \langle h_A| \delta_{I, \sigma(I')}, \quad Q_\tau := \delta_{J, \tau(J')} e^{i(E_k - E_j)t}.
\tag{12}
$$

They are determined by checking, for each $\sigma$ or $\tau$, which Kronecker-delta's are satisfied. The four possibilites are listed below.

Then,

$$
\begin{aligned}
R_{\mathrm{Id}} &= |g_A\rangle \langle h_A| \delta_{(g_A,h_B),1} \delta_{1,(h_A,h_B)} = |g_A\rangle \langle h_A| \delta_{g_A,1} \delta_{h_A,1} \delta_{h_B,1}^2 = |1_A\rangle \langle 1_A| \\
R_{(12)} &= |g_A\rangle \langle h_A| \delta_{(g_A,h_B),(h_A,h_B)} \delta_{1,1} = |g_A\rangle \langle h_A| \delta_{g_A,h_A} \delta_{h_B,h_B} = d_B \mathbb{1}_A = d \mathbb{1}_A / d_A \\
Q_{\mathrm{Id}} &= \delta_{j,j} \delta_{k,k} e^{i(E_k - E_j)t} \equiv \chi(t) \\
Q_{(12)} &= \delta_{j,k} \delta_{k,j} e^{i(E_k - E_j)t} = \delta_{j,j} \cdot 1 = d
\end{aligned}
\tag{13}
$$

---

[5] As $\rho_A$ is a matrix quantity, also these elements must in turn be expressed in bra and kets. Again here summation is implied, regardless of some indices being inside bra's/kets, and some inside the definition of the multi-indices.

where we also define auxilliary functions which will reappear numerous times,

$$\iota(t) := \sum_j e^{iE_j t} \in \mathbb{C}, \tag{14}$$

and

$$\chi(t) := \sum_{j,k=1}^{d} e^{i(E_k - E_j)t} = |\iota(t)|^2 = d + 2 \sum_{j<k} \cos((E_j - E_k)t), \tag{15}$$

which holds all the time and energy dependence of the average. The latter function will be prominent in the next section.

We can take this average in (11) as a type of inner product of vectors defined through a real symmetric[6] $q! \times q!$ matrix $\mathrm{Wg}(d, \sigma\tau^{-1})$, that only depends on $d$ and $q$.

As an example, for $q = 2$ the *Weingarten matrix* takes the form:

$$\begin{pmatrix} \mathrm{Wg}(d, \mathrm{Id}) & \mathrm{Wg}(d, (12)) \\ \mathrm{Wg}(d, (12)) & \mathrm{Wg}(d, \mathrm{Id}) \end{pmatrix} = \frac{1}{d(d^2 - 1)} \begin{pmatrix} d & -1 \\ -1 & d \end{pmatrix}. \tag{16}$$

Performing the inner product:

$$\langle \rho_A(t) \rangle = \left( \frac{\chi(t) - 1}{d^2 - 1} \right) |1_A\rangle \langle 1_A| + \left( \frac{d^2 - \chi(t)}{d^2 - 1} \right) \frac{\mathbb{1}_A}{d_A}. \tag{17}$$

The initial condition is visible in the first term and it competes with the maximally mixed state in the second term. This expression satisfies a number of consistency checks: at $t = 0$, $\chi(0) = d^2$ so $\langle \rho_A(t) \rangle$ is the inital state, and $\mathrm{Tr}_A \langle \rho_A(t) \rangle = 1 \; \forall t$, which is to be expected: integrating and tracing are linear operations and should commute, and trivially $\int dV \, \mathrm{Tr}_A \rho_A = \int 1 dV = 1$. A more general form, valid for entangled initial states, is found in [28].

## 3.2 Purity

A larger, but similar calculation can be done for an average over all eigenbases of the purity, defined in (6). For the technical details, see appendix B.

The expression for $\gamma$ is again a polynomial, so we can integrate it over the whole unitary group. In this case, as $\rho_A$ was second order in $V, V^\dagger$, $\gamma$ is fourth order, and $q = 4$. This means the vectors $R_\sigma, Q_\tau$ are 24-dimensional. However, the approach is the same and the result is strikingly compact[7]:

$$\langle \gamma(t) \rangle = \frac{\xi(t)}{d^2(d-1)(d+3)} \left( 1 - \frac{d_A + d_B}{d+1} \right) + \frac{d_A + d_B}{d+1}. \tag{18}$$

Here $\xi(t)$ is again a real function accounting for the energy and time dependence. In terms of $\iota(t)$ from (14):

$$\xi(t) := \left| \iota^2(t) + \iota(2t) \right|^2 - 4|\iota(t)|^2. \tag{19}$$

As a consistency check, indeed $\xi(0) = d^2(d-1)(d+3)$ for any spectrum, so $\langle \gamma(0) \rangle = 1$, the initial state is pure. Also (19) is symmetric in $A \leftrightarrow B$, which is to be expected: the nonzero eigenvalues of the Schmidt decompositions of $\rho_A$ and $\rho_B$ are equal, so the purities of their

---

[6]Wg only has as many unique elements as there are partitions $\phi \vdash q$, for instance, the diagonal elements are all given by $\mathrm{Wg}(\mathrm{Id}, d)$, because the Weingarten function only depends on the conjugacy class of $\sigma\tau^{-1}$.

[7]Even more expressions can be found, for instance for the matrix-valued variance of $\rho_A$. The calculation is similar in scale to that of $\gamma$, but the result is not so aesthetically pleasing.

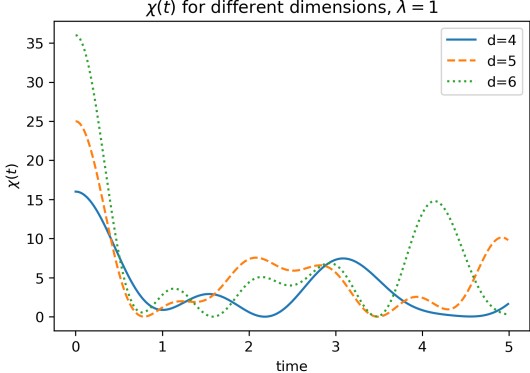

Figure 2: The function $\chi(t)$, defined in (15), for the spectrum of a Hamiltonian drawn from the $d = 4, 5, 6$ GUE, respectively.

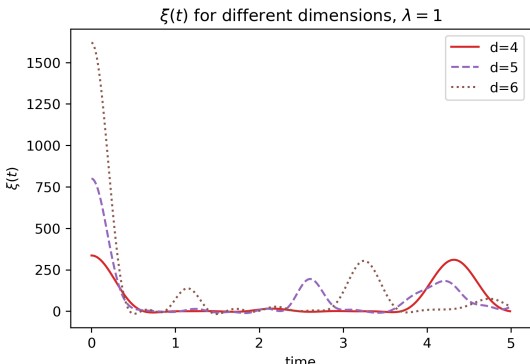

Figure 3: The function $\xi(t)$, defined in (19), for the spectrum of a Hamiltonian drawn from the $d = 4, 5, 6$ GUE, respectively.

reduced density matrices agree [42]. Also note, for a trivial bath $d_B = 1, d = d_A$, we have $\langle\gamma(t)\rangle = 1$. This makes sense: as nothing is traced out, no information is lost and the state remains pure. Combining the last two observations, a trivial subsystem ($d_A = 1$) is also always pure.

### 3.3 The Functions $\chi(t)$ and $\xi(t)$

The time and energy dependence in both main expressions of this section collect neatly into two functions, $\chi(t)$ for the density matrix, and $\xi(t)$ for the purity. These functions deserve some attention.

Though they administer the competition between the initial information remaining in $A$, and it being swept into correlations with $B$, the $\chi(t)$ and $\xi(t)$ remarkably don't depend on the partition $A + B$, only on the product $d_A d_B = d$. Examples of $\chi(t)$ and $\xi(t)$, for spectra of Hamiltonians drawn randomly from the ($\lambda = 1$) GUE are plotted in figures 2 and 3. Physically, at $t = 0$, the system is pure. This coincides with all the phasors in definitions of $\chi(t)$ and $\xi(t)$ in (15), (19) being evaluated at zero: they are in phase. A high value of $\chi(t)$ and $\xi(t)$ is thus associated with the state being pure. After some time, we expect the phasors to decohere. Then the functions drop, and we associate this with a transition to a mixed 'phase'. More quantitative statements will be made in the next sections, when eigenvalue distribution (and thus dynamics) is treated.

The spectra used to plot the figures 2 and 3 exhibit level repulsion [40], so energies are nondegenerate. Few of the oscillatory terms in $\chi, \xi$ are then coherent for long. By comparing to more erratic figures (not shown) plotted with uncorrelated energies, this level repulsion appears to be an important feature driving the rapid and sustained dying down of the functions to a stable value as $t \to \infty$.

## 4 Radial Integral: Correlators in Energy

In the previous section, we computed the average over $\mathcal{U}(d)$ of a number of expressions involving the reduced density matrix. In this section, we continue this job by also performing the weighted average over the matrix ensemble of $\chi(t)$ in (15) and $\xi(t)$ in (19). These hold the energy dependence of $\langle\rho_A(t)\rangle$ and $\langle\gamma(t)\rangle$, respectively. After this we have a full average over

the space $\mathcal{M}$ of GUE Hamiltonians. This is analogous to performing the radial part of an integral in polar coordinates. In a later section, we will also consider Poissonian (uncorrelated) energies.

In fact, these calculations take the form of 2-, 3- and 4-point exponential correlators, when we interpret the energies as the positions of the particles of a gas living in one dimension [40]. As explained in subsection A.2, we will make use of the theory of Orthogonal Polynomials.

## 4.1 Calculation of Two-Point Correlator

$\chi(t)$ in expression (15) is a sum of $d^2$ terms. The strategy will be to average them each separately, to obtain $\langle \chi(t) \rangle$. An important simplification stems from the invariance of $\chi(t)$ under exchange of any two variables $E_j, E_k$. Specifically, it consists of a sum over $j, k$ of terms $e^{i(E_k - E_j)t}$, which, upon integrating out the energies, are of course all identical as long as $k \neq j$. The $d$ remaining terms with $j = k$ are each constant unit and average to one. Let us therefore set $k = 1, j = 2$ without loss of generality:

$$\langle \chi(t) \rangle = d(d-1) \langle e^{i(E_1 - E_2)t} \rangle + d. \tag{20}$$

In appendix B, the technique of these integrals is explained, allowing us to integrate out directly all $E_j$ with $j > 2$ by introducing the symmetric kernel $K_d$. See equations (60) and (7),

$$\frac{d!}{(d-n)!} \int_{\mathbb{R}} dE_{n+1} \dots \int_{\mathbb{R}} dE_d \Delta^2(E) \mathcal{P}(E) = \det_{1 \leq j,k \leq n} \left[ K_d(E_j, E_j) \right] \tag{21}$$

we evaluate the integral leaving the natural prefactor $d(d-1)$,

$$
\begin{aligned}
d(d-1) \langle e^{i(E_1 - E_2)t} \rangle &= \int_{\mathbb{R}^2} dE_1 dE_2 \det_{1 \leq j,k \leq 2} [K_d(E_j, E_k)] e^{i(E_1 - E_2)t} \\
&= \int_{\mathbb{R}^2} dE_1 dE_2 \Big( K_d(E_1, E_1) K_d(E_2, E_2) - K_d^2(E_1, E_2) \Big) e^{i(E_1 - E_2)t} \\
&= \operatorname{Tr} F(t) \cdot \operatorname{Tr} F(-t) - \operatorname{Tr}[F(t) F(-t)].
\end{aligned}
\tag{22}
$$

Here we have defined a new symmetric matrix-valued function $F(t)$. Its elements are given for indices $0 \leq \mu, \nu \leq d-1$:

$$F_{\mu,\nu}(t) := e^{-\frac{1}{2}t^2} \sqrt{\mu! \nu!} \sum_{\alpha=0}^{\min(\mu,\nu)} \frac{(it)^{\mu+\nu-2\alpha}}{\alpha!(\mu-\alpha)!(\nu-\alpha)!}. \tag{23}$$

It may be noted that $\operatorname{Tr} F(t) = \operatorname{Tr} F(-t) = e^{-\frac{1}{2}t^2} \cdot L_{d-1}^{(1)}(t^2)$, for $L_{\mu}^{(\alpha)}$ a generalized Laguerre polynomial. The derivation is found in appendix C.

It appears $\langle \chi(t) \rangle$ is a function of $t^2$. For an example, in the simplest case[8] of two coupled qubits, the function looks like[9]:

$$\langle \chi(t) \rangle|_{d=4} = \left( 12 - 48t^2 + 46t^4 - \frac{64}{3}t^6 + \frac{25}{6}t^8 - \frac{1}{3}t^{10} \right) e^{-t^2} + 4. \tag{24}$$

We have plotted $\langle \chi(t) \rangle$ for a number of dimensions. See figure 4. The plot begins at its global maximum of $\langle \chi(0) \rangle = d^2$, drops precipitously, oscillates a number of times somewhat

---

[8]Although $\chi(t)$ can be defined for $d < 4$, it is of no use for the study of composite Hilbert spaces.

[9]From the definition of $F_{\mu,\nu}(t)$, one would expect the polynomial to be of higher degree, with leading term $t^{4d-4}$ but the highest term always cancels, for any $d$.

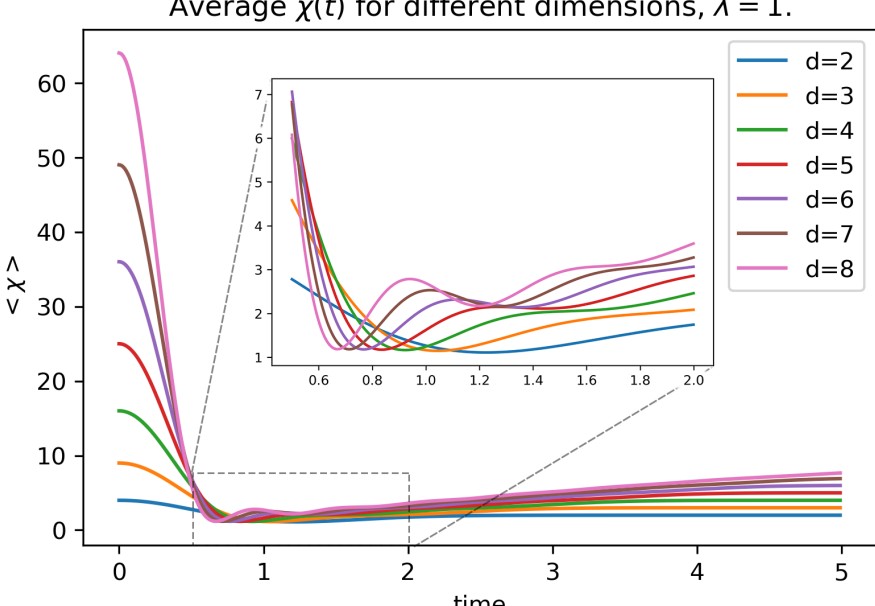

Figure 4: $\langle \chi(t) \rangle$, for $d \in \{2, 3, \ldots 8\}$.
$\lambda = 1$, the variance of the GUE used.

proportional to the dimension, and slowly *climbs* to its stable value of $d$ at infinity. The positions of the extrema satisfy $\frac{d\langle \chi \rangle}{dt} = 0$ which amounts to finding the roots of a high-dimensional polynomial. This is done via Powell's method. Solving for the position of the global minimum, the first local minimum, it appears more quickly as $d$ increases and neatly follows the fit $t_{\min} \approx 1.93/\sqrt{d + 0.45}$, found on $d \in [2, \ldots, 84]$. Despite the initial maximum scaling as $d^2$, the global minimum converges from below to a *constant* as we increase $d$, empirically: $\lim_{d \to \infty} \langle \chi(t_{\min}) \rangle \approx 1.908$.

These techniques readily generalize to larger correlators.

## 4.2 Calculation of Higher Correlators

Continuing the approach above, we will evaluate the integrals needed to find $\langle \xi(t) \rangle$. As a more general application, seen as a field theory of an eigenvalue gas [43], these techniques allow one to calculate vertex operators and $n$-point correlators in the GUE distribution.

All terms in $\xi(t)$ are of the form $e^{i(E_k + E_m - E_j - E_l)t}$, however some $j, k, l, m$ may coincide. Again, after it has been determined which indices are distinct, the actual value of each index is irrelevant for the expectation value. A different correlator will appear for choosing $k = m$ than for $k = j$ etc., due to signs. Upon careful inspection of expression (19), we decompose

$$
\begin{aligned}
\langle \xi(t) \rangle = {} & 4d(d-1)\left\langle e^{2i(E_1 - E_2)t} \right\rangle + \frac{2d!}{(d-3)!}\left\langle e^{i(2E_1 - E_2 - E_3)t} \right\rangle + \frac{2d!}{(d-3)!}\left\langle e^{i(E_1 + E_2 - 2E_3)t} \right\rangle \\
& + \frac{d!}{(d-4)!}\left\langle e^{i(E_1 + E_2 - E_3 - E_4)t} \right\rangle + 4d(d-1)^2\left\langle e^{i(E_1 - E_2)t} \right\rangle + 2d(d-1).
\end{aligned}
\tag{25}
$$

We recognize the two-point correlator from the previous section in the last pair of brackets. Also, the first term is simply the same correlator with double time. What remains are the three- and four-point correlators. In fact it will turn out that all these expressions are real, and as both three-point correlators are each others complex conjugate, they are also equal. Rather than

subsituting everything into a final, cumbersome expression, we will explain how to evaluate general $n$-point functions, with more detail in appendix C.

The first characterization is in terms of an exponential generating function $G(\{a_m\})$ of dummy variables $a_m$, enumerated by the possible phase multiples $m \in \mathbb{Z}$. Formally:

$$G(\{a_m\}) := \det_{d \times d}\left(\mathbb{1} + \sum_{m \in \mathbb{Z}} a_m F(mt)\right). \tag{26}$$

Indeed this determinant of a symmetric matrix is always real, and therefore also the resulting correlators. For any $n$-point correlator characterized by integers $\{c_1, \ldots, c_n\}$:

$$\frac{d!}{(d-n)!}\left\langle \prod_{j=1}^{n} e^{ic_j E_j t}\right\rangle = \left(\prod_{j=1}^{n} \frac{\partial}{\partial a_{c_j}}\right) G(\{a_m\})\bigg|_{\{a_m\}=\{0\}}. \tag{27}$$

This coincides, in index notation, with the following[10]:

$$\frac{d!}{(d-n)!}\left\langle \prod_{j=1}^{n} e^{ic_j E_j t}\right\rangle = \sum_{\sigma \in S_n} \sum_{v_1, v_2, \ldots v_n=0}^{d-1} \mathrm{sgn}(\sigma) \prod_{j=1}^{n} F_{v_j, v_{\sigma(j)}}(c_j t)$$
$$= \sum_{v_1, v_2, \ldots v_n=0}^{d-1} \det_{1 \le j,k \le n} F_{v_j, v_k}(c_j t). \tag{28}$$

Going from expression (21) to (28), we expand the $(n \times n)$ determinant using the Leibniz formula into a sum over permutations $\sigma \in S_n$. In order to interpret this result, it is best to try an example. For instance, in the case of $d(d-1)(d-2)\langle e^{i(2E_1-E_2-E_3)t}\rangle$, $n = 3$ and $(c_1, c_2, c_3) = (2, -1, -1)$. The term corresponding to $\sigma = \mathrm{Id}$, with sign $+1$, will result in tracing each instance of $F$ with itself: $\mathrm{Tr}\,F(2t) \cdot \mathrm{Tr}\,F(-t) \cdot \mathrm{Tr}\,F(-t)$. The term in $\sigma = (12)$ will couple the first index to the second, the second to the first, and the third to itself: $-\mathrm{Tr}[F(2t)F(-t)] \cdot \mathrm{Tr}\,F(-t)$, just as $\sigma = (13)$. $\sigma = (123)$ contributes $\mathrm{Tr}[F(2t)F(-t)F(-t)]$, etc. All these contractions can be found by expanding the determinant in expression (27) as $\det = \exp \circ \mathrm{Tr} \circ \log$ and collecting the suitable factors of $a_m$. In the current example, we would be interested in any and all terms multiplied by exactly $a_2 a_{-1}^2$. Replacing $a_2 a_{-1}^2 \to 2$ due to the derivative $\partial_{a_{c_{-1}}}$ results in the answer. Take note that from $n = 4$, the order inside the trace will begin to matter.

$$\frac{d!}{(d-3)!}\left\langle e^{i(2E_1-E_2-E_3)t}\right\rangle = \mathrm{Tr}\,F(2t) \cdot (\mathrm{Tr}\,F(-t))^2 - 2\,\mathrm{Tr}[F(2t)F(-t)] \cdot \mathrm{Tr}\,F(-t)$$
$$- \mathrm{Tr}\,F(2t)\,\mathrm{Tr}[F^2(-t)] + 2\,\mathrm{Tr}[F(2t)F(-t)F(-t)]. \tag{29}$$

The explicit forms of all needed correlators are found in appendix C. Again, $\langle \xi(t)\rangle$ will take the form of polynomials times exponents of $t^2$. For instance, for $d = 4$, or 2 qubits, it

---

[10] The results of the correlators were confirmed by straightforward integration of equations (22), (29) and (106) for Hilbert spaces up to $d = 8$.

works out to

$$
\begin{aligned}
\langle \xi(t) \rangle |_{d=4} = 24 &+ \left( 144 - 576t^2 + 552t^4 - 256t^6 + 50t^8 - 4t^{10} \right) e^{-t^2} \\
&+ \left( 24 - 192t^2 + 448t^4 - \frac{1024}{3}t^6 + \frac{256}{3}t^8 \right) e^{-2t^2} \\
&+ \left( 96 - 1152t^2 + 3312t^4 - 3328t^6 + 1548t^8 - 216t^{10} \right) e^{-3t^2} \\
&+ \left( 48 - 768t^2 + 2944t^4 - \frac{16384}{3}t^6 + \frac{12800}{3}t^8 - \frac{4096}{3}t^{10} \right) e^{-4t^2}.
\end{aligned}
\tag{30}
$$

For $d = 4, 5, 6$ we have plotted the shape of $\langle \xi(t) \rangle$, in figure 5. It is clear that the function begins at its global maximum $\langle \xi(0) \rangle = d^2(d-1)(d+3)$, drops close to zero, and slowly climbs to its steady state value of $2d(d-1)$. The positions of the global minima also follow an inverse square root fit, with slightly different constants than in the $\langle \chi(t) \rangle$ case. We find, on $d \in [4, \ldots, 20]$: $t_{\min} \approx 1.95/\sqrt{d + 1.69}$ minimizes $\langle \xi(t) \rangle$.

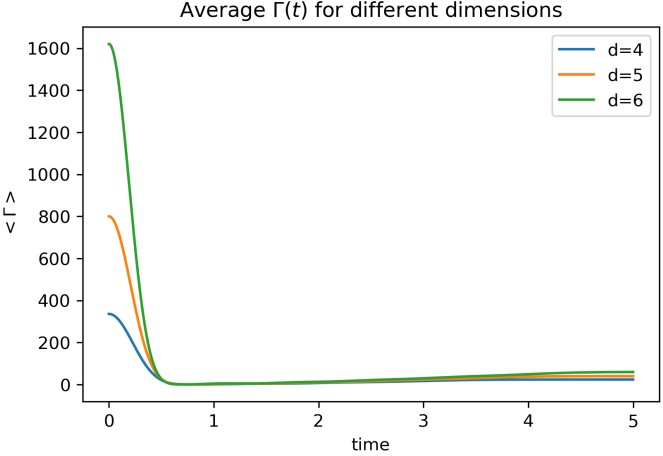

Figure 5: $\langle \xi(t) \rangle$ for $d = 3, 4, 5$

# 5 Final Results and Discussion

Here we state the full expressions for the two main results of this paper, and discuss their properties.

## 5.1 Dynamical GUE-averaged Reduced Density Matrix

The first result is the time-dependent, GUE average reduced density matrix of subsystem $A$. When a bipartite system $A + B$ is coupled by a $\lambda = 1$ GUE Hamiltonian, the average local state of $A$ is given by:

$$
\langle\langle \rho_A(t) \rangle\rangle = \left( \frac{\langle \chi(t) \rangle - 1}{d^2 - 1} \right) |1_A\rangle \langle 1_A| + \left( \frac{d^2 - \langle \chi(t) \rangle}{d^2 - 1} \right) \frac{\mathbb{1}_A}{d_A},
\tag{31}
$$

with

$$
\langle \chi(t) \rangle = (\operatorname{Tr} F(t))^2 - \operatorname{Tr}[F(t)F(-t)] + d.
\tag{32}
$$

The definition of the the matrix $F(t)$ is given in (23). This can be thought of as a GUE-induced, time-dependent ensemble on the basis states of $\mathcal{H}_A$. It satisfies some consistency checks, for instance $\text{Tr}\langle\langle\rho_A(t)\rangle\rangle = 1$, as it should: the integrand always has unit trace. Also, $\langle\langle\rho_A(0)\rangle\rangle = |1_A\rangle\langle 1_A|$, before time-evolution, the average is always in the initial state. Also, as it appears $\langle\chi(t)\rangle > 0$, it is easy to see $\langle\langle\rho_A(t)\rangle\rangle > 0$: it is positive semi-definite and is thus a well defined density matrix [41]. Besides the normalization (to unit trace) of the projectors, it is remarkable that this expression does not depend on the partition of $(d_A, d_B)$, only on their product.

It is interesting to visualize the competition between the projectors $|1_A\rangle\langle 1_A|$ and $\mathbb{1}_A/d_A$ in equation (31). In order to be specific, we will consider coupling a single qubit as subsystem $A$, to a bath with $\mathcal{H}_B$ of $d_B = 2, 3, 4$. See figure 6. As the bath size increases, indeed the mixed component becomes more dominant. We pointed out in the previous sections that $\chi(t)$ administers the competition between the pure and mixed states of $A$. When the phases that comprise $\chi(t)$ in (15) are coherent, $\chi(t)$ is large and $A$ is pure. As they decohere, $A$ mixes. We observe that shortly ($t_{\min} \approx 2/\sqrt{d}$) following coupling $A$ to $B$, the mixing becomes approximately complete. After this dip, the initial condition $|1_A\rangle\langle 1_A|$ resurfaces and then stabilizes to a degree. Initial information disappears rapidly, then trickles back in. At high $d$, the coefficient of $|1_A\rangle\langle 1_A|$ at $t_{\min}$ falls off as $0.908/d^2$. However, the late time limit is

$$\lim_{t\to\infty}\langle\langle\rho_A(t)\rangle\rangle = \left(\frac{1}{d+1}\right)|1_A\rangle\langle 1_A| + \left(\frac{d}{d+1}\right)\frac{\mathbb{1}_A}{d_A}. \tag{33}$$

Also taking $d_B \to \infty$, the mixing becomes complete instantaneously. This is to be expected: increasing the degrees of freedom of the bath without decreasing the interaction to each of them.

It is almost futile to contrast the analytic results of figure 6 with numerical simulations, so good is the agreement. See figures 7 and 8, in which three times $N = 10000$ randomly generated GUE ($\lambda = 1$) Hamiltonians were used to couple a qubit to the same three baths as in the analytic example. The initial product state was drawn randomly according to the Haar measure from $\mathcal{H}_A$ and $\mathcal{H}_B$. For each system, $\rho_A(t)$ was calculated, and then averaged over the set of $N$. Of this average, the $|1_A\rangle\langle 1_A|$ occupation is plotted, as well as that of the other state[11], $|2_A\rangle\langle 2_A|$. To an accuracy $\approx 1/N$, the cross-components vanish.

Using the same techniques, other polynomial functions of the reduced density matrix can be averaged over the GUE. We have simply treated the most obvious candidates. On that note, by linearity, the GUE-average of the expectation value $\bar{o}(t)$ of any constant observable $O_A$ on $A$ can immediately be found from the average reduced density matrix. Integrating and tracing are linear operators, and thus commute,

$$\langle\langle\bar{o}(t)\rangle\rangle = \int_{\mathcal{M}}\mathcal{P}(H)dH\,\text{Tr}_A(O_A\rho_A(t))$$
$$= \text{Tr}_A\left(O_A\int_{\mathcal{M}}\mathcal{P}(H)dH\rho_A(t)\right) = \text{Tr}_A(O_A\langle\langle\rho_A(t)\rangle\rangle). \tag{34}$$

The matrix-valued variance of the density matrix has also been averaged over the unitary group. This is nowhere zero, as expected, with smaller elements along the diagonal and a larger ones on the first row and column. In the large $d$ limit, all go to zero at least at the same rate as the density matrix elements. This adds credence to the average. However, the explicit expression is cumbersome and not very insightful, so it has been omitted from this work.

---

[11]The difference in range of the plots is due to the normalization of the projectors, and the fact that element occupations are the sum and difference of the projectors coefficients. In numerics, we have no access to the exact decomposition of equation (31).

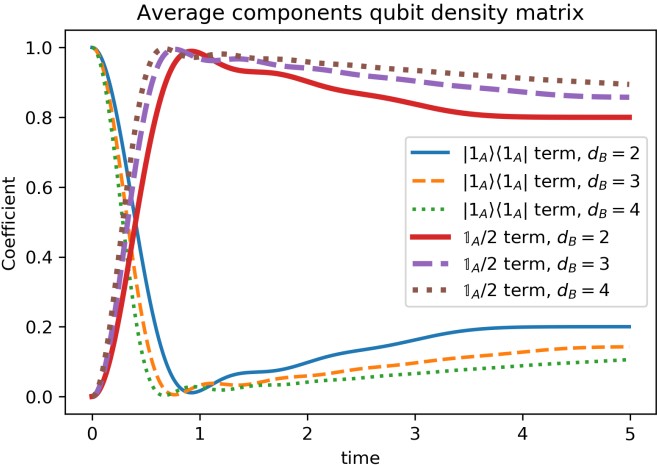

Figure 6: coefficients of $|1_A\rangle\langle 1_A|$ vs $\mathbb{1}_A/d_A$ in the $\langle\langle\rho_A(t)\rangle\rangle$ of a qubit, coupled to a bath of $d_B \in \{2,3,4\}$.

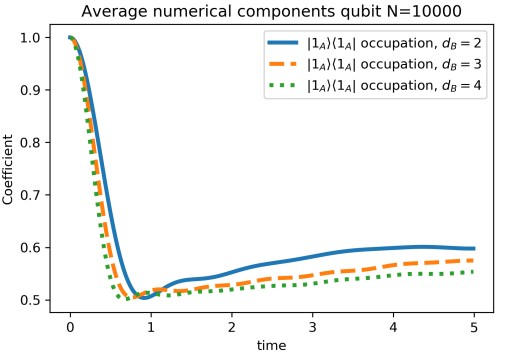

Figure 7: Numerically GUE-averaged coefficients of $|1_A\rangle\langle 1_A|$ in $\rho_A(t)$ of a qubit, coupled to a bath of $d_B \in \{2,3,4\}$.

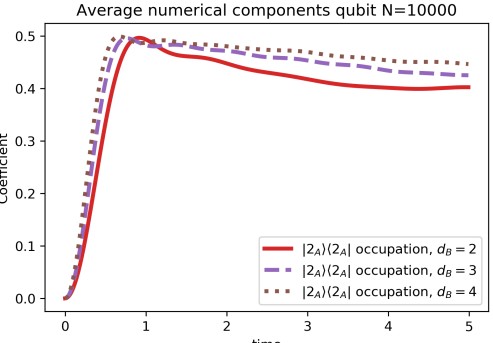

Figure 8: Numerically GUE-averaged coefficients of $|2_A\rangle\langle 2_A|$ in $\rho_A(t)$ of a qubit, coupled to a bath of $d_B \in \{2,3,4\}$.

## 5.2 Dynamical GUE-averaged Subsystem Purity

The second result of this paper is the Dynamical GUE-average subsystem purity. It is the average purity of $A$ entangled to $B$ under collective evolution of a $\lambda = 1$ GUE Hamiltonian.

$$\langle\langle\gamma(t)\rangle\rangle = \frac{\langle\xi(t)\rangle}{d^2(d-1)(d+3)}\left(1 - \frac{d_A + d_B}{d+1}\right) + \frac{d_A + d_B}{d+1}, \tag{35}$$

where $\langle\xi(t)\rangle$ is defined in (25). This expression also satisfies a number of consistency checks: $\langle\langle\gamma(0)\rangle\rangle = 1$ by the setup, and also if $d_A = 1$ or $d_B = 1$, $\langle\langle\gamma(t)\rangle\rangle = 1$, as a trivial system is always pure and a trivial bath cannot entangle. In agreement with the behavior of $\langle\langle\rho_A(t)\rangle\rangle$, the purity quickly moves from a pure state to its most mixed value, and then recovers somewhat slowly. We can visualize expression (35), for qubit and qutrit subsystems $A$ coupled to varying baths. See figures 9 and 10.

Numerics of the same strain again confirm these plots. For figure 11 in the case $(d_A, d_B) = (2,2)$ and figure 12 in the case $(d_A, d_B) = (2,3)$, sets of Hamiltonians are sampled from the GUE. For each $H$, the subsystem purity was calculated over time. Sample size varied between 1, 10, 100 and 1000 Hamiltonians. Then for each time, this value was averaged over the set.

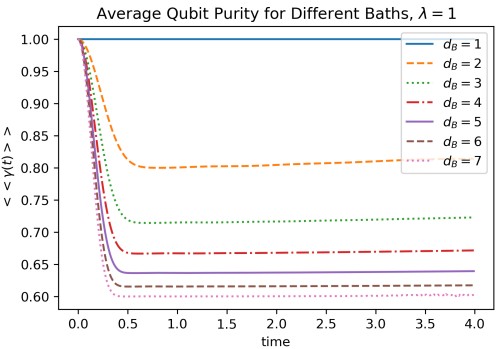

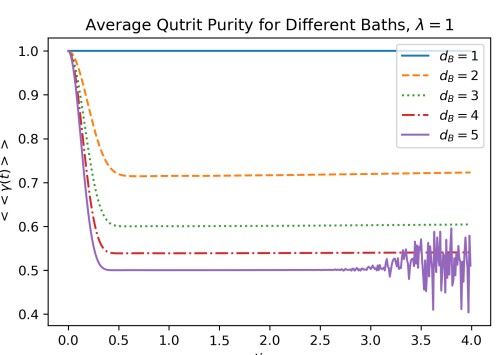

Figure 9: Dynamical GUE-averaged Subsystem purity $\langle\langle\gamma(t)\rangle\rangle$ of a qubit, coupled to baths of $d_B \in [1,\dots,7]$. The GUE variance $\lambda = 1$.

Figure 10: $\langle\langle\gamma(t)\rangle\rangle$ of a qutrit, coupled to baths of $d_B \in [1,\dots,5]$. $d_B = 5$ involves polynomials of degree up to 98 in $t$, admitting numerical error.

The result shows a convincing convergence to our calculated average.

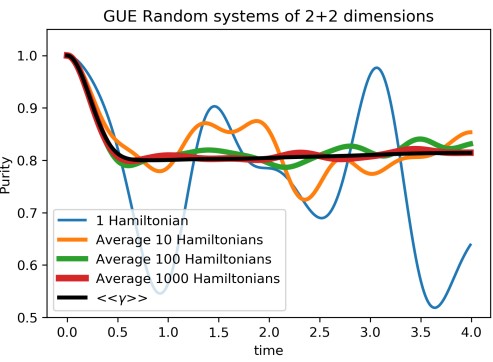

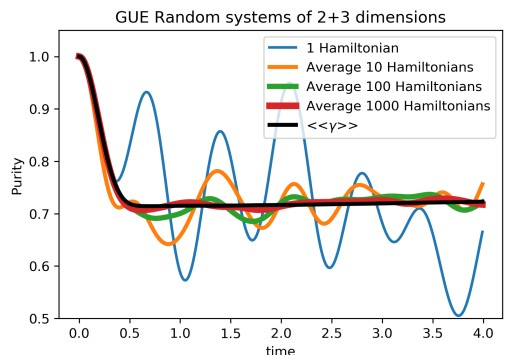

Figure 11: Numerical average purity: a qubit-qubit system for growing samples of GUE-Hamiltonians. In black the analytic $\langle\langle\gamma(t)\rangle\rangle$.

Figure 12: Numerical average purity: a qubit-qutrit system for growing samples of GUE-Hamiltonians. In black the analytic $\langle\langle\gamma(t)\rangle\rangle$.

The late time behaviour is readily given. Remember that in expression (25), the exponents will eventually shrink to zero, their prefactors are polynomial

$$\lim_{t\to\infty} \langle\langle\gamma(t)\rangle\rangle = \frac{2}{d(d+3)}\left(1 - \frac{d_A + d_B}{d+1}\right) + \frac{d_A + d_B}{d+1}. \tag{36}$$

The plots teach us convergence to this limit is strong and uniform. Compare this to the *trace measure* average purity obtained by drawing a random vector from a $d = d_B \cdot d_A$ dimensional Hilbert space according to the Haar measure, and tracing out the $d_B$ dimensions of the bath [44],

$$\langle\bar{\gamma}\rangle_{d_A,d_B} = \frac{d_A + d_B}{d+1}. \tag{37}$$

The limit of the dynamical case, (36), exhibits an extra positive term: a remnant of the initial purity. Note that there has been much research into such averages in RMT, results include the Von Neumann and Renyi entropies of such trace measures, which are more general or powerful descriptors, also at infinite dimensional Hilbert spaces [45,46]. What distinguishes our work, is that the subsystem is found by *evolving* according to the random matrix as a

Hamiltonian, instead of the full system state being uniformly random, which would result in $\rho_A$ being drawn directly from a matrix ensemble.

## 5.3 Oscillations: Comparison to Poissonian Ensembles

We will take a moment to spotlight a peculiar feature of our dynamical averages. They exhibit oscillations, as shown in figures 4 and 6. We may count oscillations by the number of extrema. For $\langle \chi(t) \rangle$, containing the dynamics of $\langle \langle \rho_A(t) \rangle \rangle$, the time at which extrema occur is plotted against $d$. See figure 13. Note that they all have a maximum at $t = 0$, which is omitted. Until $d = 5$, there is just one minimum, and in steps a new 'fold' with a maximum and minimum is added.

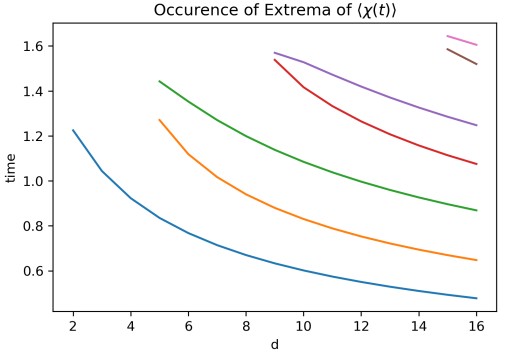

Figure 13: Plot of extremum positions of $\langle \chi(t) \rangle$ against $d$. until $d = 5$, there is just one minimum, in steps new 'folds' with a max and min appear.

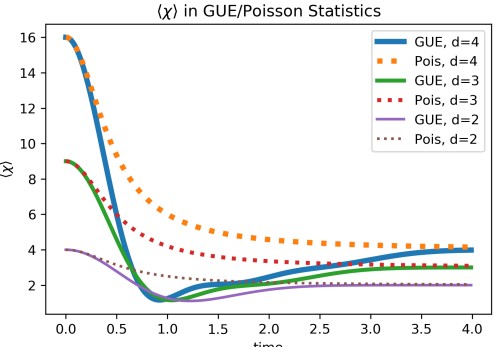

Figure 14: Comparison of $\langle \chi(t) \rangle$ to $\langle \chi(t) \rangle_P$, the former over the GUE and the latter over Poisson energy statistics.

Each individual Hamiltonian will drive eternal oscillations, and there is always Poincaré recurrence. Their averages however are characterized by finite dimensional Laguerre polynomials, which form a landscape with a finite amount of 'features'. They will settle down eventually. This allows us to classify a period of *pre-equilibration* rigorously, as the period between the first and last extremum. After this, the system is on its way to equilibrium. This plot also portrays characteristic frequencies in the period of pre-equilibration, which stem from the Gaussian distribution of the energies. The appearance of the Laguerre polynomials can be traced back to the Vandermonde determinant: the repulsion of eigenvalues giving rise to *Wigner-Dyson* statistics, causes the phases to decohere in a specific way. To illustrate, consider a a very artificial ensemble where the eigenstates are still uniform in the Haar measure, but we replace the distribution of energy differences such that all levels are decoupled. This results in *Poisson* statistics between neighboring levels. Such level spacing is seen to describe integrable systems more accurately. Specifically, all the above calculations can be repeated with factorized eigenvalue distribution

$$P(E) = \prod_{j=1}^{d} \left( \mu e^{-\mu E_j} \right) = \mu^d \exp\left( -\mu \sum_j E_j \right), \tag{38}$$

and then each absolute energy gap $|E_1 - E_2|$ will also follow an exponential distribution[12], which is the hallmark of Poisson statistics for neighboring levels.

---

[12]The exponential distribution describes a random variable measuring the interval between instances of a Poisson process. This can be thought of as the waiting time before a random event with a constant probability. The difference of two such variables is then the time between two independent events. Assume an ordering, without loss of generality. After the first has occurred, the distribution of the time until the second is independent of

We may thus average $\chi(t)$ and even $\xi(t)$ with respect to this, taking care to scale the distribution sensibly with dimension. The considerably simpler calculation can be found in Appendix D. The result is

$$\langle \chi(t) \rangle_P = d + \frac{d(d-1)}{(d+1)t^2+1}. \tag{39}$$

This function has the same limiting behavior, for $t = 0$ and $t \to \infty$, but at intermediate times, decays more slowly, and exhibits no oscillations. See figure 14. In integrable Systems the degrees of freedom are more independent, or are thought to be less highly coupled, and do not exhibit energy level repulsion. Instead the largest probability occurs at zero gap. [40]. In such systems, we expect entanglement to grow more slowly.

## 5.4   Bessel Function Scaling Limit

In a related work, discovered after most of our computations had been completed, we encountered a certain scaling limit of what is essentially the upper left coefficient in $\langle\langle \rho_A(t) \rangle\rangle$ in expression (31) [28]. This function does not diverge or vanish as $d \to \infty$, so long as we take care to scale time by a factor $\sqrt{d}$, equivalent to setting $\lambda = d$ in the GUE distribution, decreasing interaction strength as we increase dimension. This agrees with the square root behaviour of the first minima. In a new time coordinate,

$$\lim_{d\to\infty} \frac{\langle \chi(\tau) \rangle - 1}{d^2 - 1} = \left| \lim_{d\to\infty} \frac{\left\langle \sum_j e^{iE_j\tau} \right\rangle}{d} \right|^2 = \left( \frac{J_1(2\tau)}{\tau} \right)^2, \quad \tau := \sqrt{d} \cdot t. \tag{40}$$

$J_1$ is a Bessel function of the first kind. Additionally, we infer the limit of what we will call the 'Arbiter' of the purity: the time dependent factor that mediates between the trace average purity and its complement in (35)

$$\lim_{d\to\infty} \frac{\langle \xi(\tau) \rangle}{d^2(d-1)(d+3)} = \left| \lim_{d\to\infty} \frac{\left\langle \sum_j e^{iE_j\tau} \right\rangle}{d} \right|^4 = \left( \frac{J_1(2\tau)}{\tau} \right)^4, \quad \tau := \sqrt{d} \cdot t. \tag{41}$$

Deviations from the limit are of order $1/d$. The convergence is apparent in figures 15 and 16.

These results were found by Fourier transforming the $d = \infty$ eigenvalue distribution, which is characterized by the *semicircle law* [40]. This invites a conjecture: after tentative calculations of $\langle \text{Tr}\, \rho_A^n \rangle$ for $n = 1, 2, 3$, we see that in the limit $d_A, d_B \to \infty$, $d_A/d_B$ fixed, the only term that does not scale to zero is $(\chi(t)/d^2)^n$, with the exception of $n = 1$, where due to normalization the trace is unit. Therefore we expect that $\langle \text{Tr}\, \rho_A^n(\tau) \rangle = (J_1(2t)/t)^{2n}$ up to $\mathcal{O}(1/d)$ corrections, for $n > 1$. This could be useful for series expansions. It has been mentioned that this structure is reminiscent of the planar limit of diagrams contributing to matrix model field theories. As this work is not carried out in a diagrammatic language, it is hard to be certain, but what is certain is that it would be worthwhile to explore the connections these theories as well as to other random matrix ensembles, for instance the quantum Rydberg blockaded/Fibonacci chain [47, 48].

---

the past, and is again exponentially distributed. It is true that the energies themselves are artificially all positive, however that is of no concern as in this work, and generally in physics, we are only interested in energy differences.

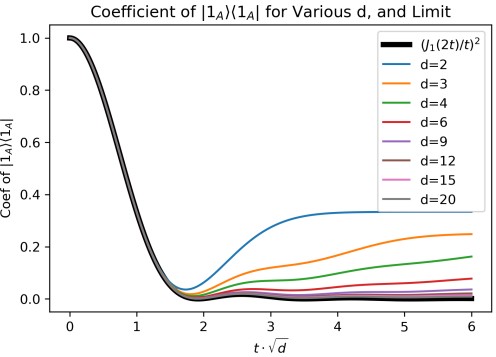

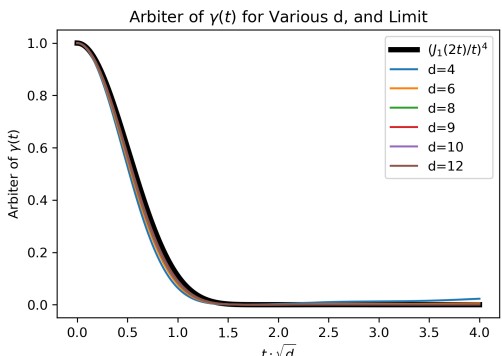

Figure 15: Simultaneous plot of the first coefficient of $\langle\langle\rho_A(t)\rangle\rangle$ for various $d$ against scaled time. Also the limit to which they converge, in black.

Figure 16: Simultaneous plot the arbiter of $\langle\langle\gamma(t)\rangle\rangle$ for various $d$ against scaled time. Also the limit to which they converge, in black.

## 5.5 Numerical Comparison to Physical Models

The philosophy of this subsection is the following: we would like to compare entanglement generation in our random matrix ensembles to *ensembles* of more physical models. Each element of the latter type of ensemble is a well studied model, but it is characterized by unique coordinates and coupling strengths. This way, we may hope their statistical moments equilibrate to a constant state over time, while each individual instance of the model obviously does not, but instead oscillates forever. This is necessary in order to compare to the GUE and Poissonian ensembles, which show this behavior. To this end, we will need to introduce randomness. We begin from the Transverse Field Ising Model (TFIM), Transverse Field XXZ-chain (XXZ), Spin Glass (SG), Central spin (CS) and Sachdev-Ye-Kitaev (SYK), because they are all integrable in the certain implementations. Contrary to what is common practice, we will be looking at (exceedingly) low dimensional versions of these models, as that is the focus of the article in general.

For each integrable ensemble, we construct a 'Disordered twin', where the local interactions are the same, but they are no longer globally coordinated, breaking integrability and introducing more randomness. The models all feature couplings $h, g$ which are drawn from standard normal distributions, and are in places scaled by external parameters labeled by $J, B \in R$. More randomness is introduced by rotating the set of axes in the TFIM and its disordered twin, the DTFIM, and the XXZ and its own disordered twin, DXXZ. This is achieved by choosing random elements $\mathcal{X}$ from the 3-dimensional representation of $SO(3)$, and using the first column of this as the new x-axis, etc. Exact implementations are below.

$$H_{\text{TFIM}}(J) = \sum_{j,a}\left(\sum_b \sigma_j^a\sigma_{j+1}^b\mathcal{X}^{1a}\mathcal{X}^{1b} + Jg\mathcal{X}^{3a}\sigma_j^a\right) \tag{42}$$

$$H_{\text{DTFIM}}(J) = \sum_{j,a}\left(\sum_b \sigma_j^a\sigma_{j+1}^b\mathcal{X}_j^{1a}\mathcal{X}_j^{1b} + Jg_j\mathcal{Y}_j^{3a}\sigma_j^a\right) \tag{43}$$

$$H_{\text{XXZ}}(B,J) = \sum_{j,a}\left(\sum_b\left(\sigma_j^a\sigma_{j+1}^b\mathcal{X}^{1a}\mathcal{X}^{1b} + \sigma_j^a\sigma_{j+1}^b\mathcal{X}^{2a}\mathcal{X}^{2b} + Jg\sigma_j^a\sigma_{j+1}^b\mathcal{X}^{3a}\mathcal{X}^{3b}\right) + Bh\mathcal{X}^{3a}\sigma_j^a\right) \tag{44}$$

$$H_{\text{DXXZ}}(B,J) = \sum_{j,a} \Big( \sum_b \big( \sigma_j^a \sigma_{j+1}^b \mathcal{X}_j^{1a} \mathcal{X}_j^{1b} + \sigma_j^a \sigma_{j+1}^b \mathcal{X}_j^{2a} \mathcal{X}_j^{2b} + J g_j \sigma_j^a \sigma_{j+1}^b \mathcal{X}_j^{3a} \mathcal{X}_j^{3b} \big) + B h_j \mathcal{Y}_j^{3a} \sigma_j^a \Big). \tag{45}$$

When stochastic variables are given a subscript $j$, it means that a new element is drawn for each site $j$, inside one Hamiltonian of the ensemble. As always, the $\sigma^a$, $a = 1, 2, 3$, are Pauli spin matrices, and $j$ runs over the sites. Moving on, $f$ are spinless Majorana Fermions. They are implemented by Jordan-Wigner transforming Pauli matrices, in a one-to-two mapping [49]. We use them in the SYK model[13]

$$H_{\text{SYK}}(J_1, J_2) = \sum_{i<j<k<l} J_2 g_{i,j,k,l} f_i f_j f_k f_l. \tag{46}$$

The Spin glass model used, is given by:

$$H_{\text{SG}}(J_1, J_2, J_3) = \sum_{j,a,b} \Big( h^a \delta_{a,b} + J_1 h^{a,b} + J_2 h_j^{a,b} \Big) \sigma_j^a \sigma_{j+1}^b + \sum_{j,a} \Big( g^a + J_3 g_j^a \Big) \sigma_j^a. \tag{47}$$

Finally, we have a central spin model, in an implementation allowing multiple central spins. [50]. There is a random local magnetic field along the 3$^{\text{rd}}$ axis, random strength $J$ coupling inside the $s_A$ number of central spins which will always coincide with system $A$, and unit random coupling between the central and $s_B$ number of bath spins, which have no other interactions, and form system $B$. This system is

$$H_{\text{CS}}(B,J) = B \sum_{j=1}^{s_A} g_j \sigma_j^3 + J \sum_{j,k=1}^{s_A} \sum_a h_{j,k}^a \sigma_j^a \sigma_k^a + \sum_{j=1}^{s_A} \sum_{k=s_A+1}^{s_A+s_B} \sum_a h_{j,k}^a \sigma_j^a \sigma_k^a. \tag{48}$$

For an example of the gap statistics, a common litmus test for the nature of the ensemble, see figure 17. Pictured are the subsequent energy gaps for the XXZ/DXXZ, as well as SYK and the GUE for comparison. It is clear that the introduced disorder has successfully broken the Poissonian character of the XXZ, as its spectrum has changed to more resemble that of the GUE. For other ensembles, the results follow the same trend from the original to the disordered. The lower graph is the ratio of subsequent gaps, a naturally scale-invariant and robust measure for the same distinction [51]. In the case of the CS model, the spectrum appears integrable for $B = 0$, and chaotic otherwise.

Wherever applicable, periodic boundary conditions apply: $j$ is taken modulo $s$, the number of sites, each having a 2-dimensional Hilbert space. In order to compare time evolution quantitatively, after the ensemble is sampled, the energies of each system are scaled such that the ensemble has numerical average $\langle (E_i - E_j)^2 \rangle = 2(d+1), i \neq j$. The system is initialized in a tensor product of a Haar-measure random state from $\mathcal{H}_A$ and one from $\mathcal{H}_B$. Then time evolution begins, according to $H$, and at each time, $\rho_A$ is computed, and finally its elements are averaged over 10000 samples from each ensemble.

As stated, the random variables are drawn from the following distributions:

$$\mathcal{X}_\bullet, \mathcal{Y}_\bullet \in_R SO(3), \tag{49}$$

uniformly according to the Haar measure, and

$$g_\bullet^\bullet, h_\bullet^\bullet \in_R \mathcal{N}(0,1). \tag{50}$$

---

[13]The order of the fermionic operator doesn't matter: reordering would simply introduce minus signs, but the prefactor $g_\bullet$ is symmetric around zero.

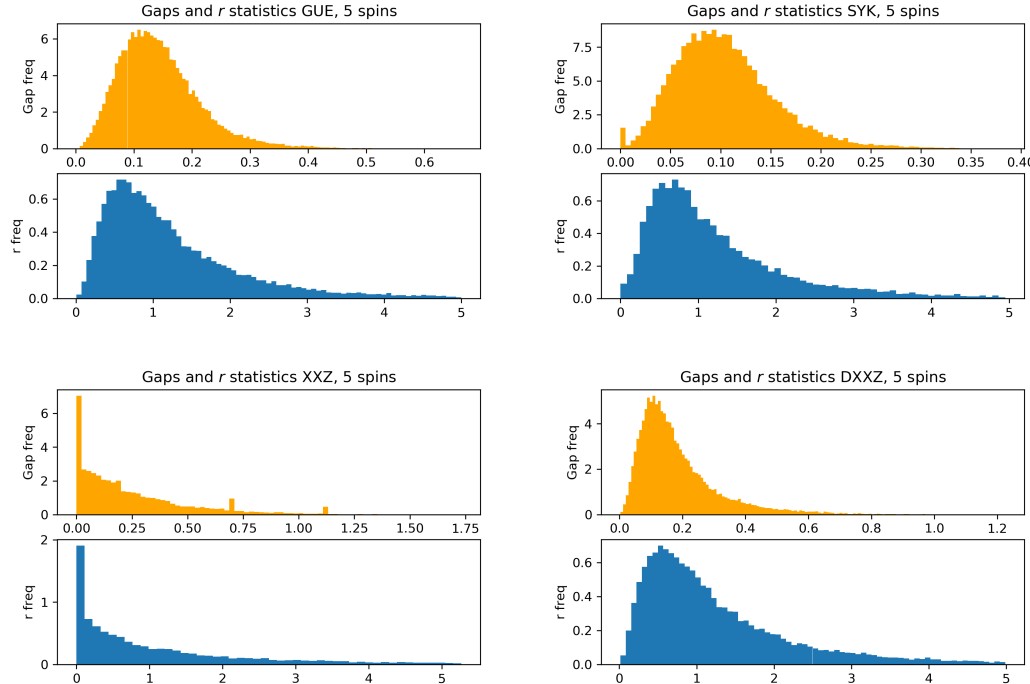

Figure 17: Distribution of energy gaps in various ensembles, along with subsequent gap ratio. Of the four pictured, the XXZ has the Poissonian character of an integrable model, the other three appear more Wigner-Dyson. XXZ is taken here with $J = B = 1$.

For each of these systems, we compare their evolution to the GUE and Poisson ensembles, described earlier. See figures 18. This allows us to get a sense of how well these mathematical predictions work for physical models. All systems initially entangle quadratically[14]. There seems to be a general trend that the disordered versions of each ensemble have more mixing, and conversely some integrable systems do not seem to equilibrate at all, which is to be expected. For example, the $J = 0$ TFIM is even free, characterized by perfect harmonic occupation.

We hypothesize the reason the physical systems often do not adhere to the predictions of either average more faithfully, is that the requirement of unitary invariance of the basis is very strong, even for the most exotic ensembles of systems. When the eigenbases of the integrable ensembles are 'scrambled', by multiplying them with a random unitary from $\mathcal{U}(d)$, their behavior very strongly resembles the Poissonian curves in figure 18 (Not shown). Finally, these systems still exhibit a number of symmetries leading to exact degeneracies, which may further influence the results. Notably, the SYK model is quite faithful to the GUE in many ways, hinting at a deeper connection. Such ties have been investigated, see e.g. [52].

In order to quantify the resemblance of the dynamics to either the exponential or the GUE ensembles, we have devised a rather simple scheme. We define the distance between dynamics of some model $M$ and the GUE as

$$D_{M,\text{GUE}}^{(6)} = \int_0^6 dt ||\rho_M(t) - \rho_{\text{GUE}}(t)||,$$
(51)

and similar for Poisson. Here $\rho_\bullet(t)$ is shorthand for the reduced density matrix of subsystem

---

[14]This is a consequence of linearity of the Schrödinger equation. The occupation of non-initial states can only grow linearly, and the corresponding density matrix element ( probability) grows quadratically, evocative of the quantum Zeno effect.

Figure 18: Average coefficient of element $|1_A\rangle\langle 1_A|$ in $\rho_A(t)$ over different model ensembles. In thick lines, also the GUE and Poissonian predictions. How well they agree varies wildly per model.

$A$ after tracing out $B$ under the specified Hamiltonian Ensemble, and the norm is the regular matrix 2-norm. This is admittedly a crude measure, but it fits with the notion that lines coinciding in figures such 18, should give a distance of zero, and large variations are punished in a squared manner. The cutoff in time of 6 is purely by eye: this captures the initial descent and a large part of the equilibration period. We take into account that there is always statistical noise as we can only sample 4000 instances from each ensemble. (Specifically, the SYK model is computationally heavy), by reporting here on the ratio of $D_{M,\text{GUE}}^{(6)}/D_{\text{GUE},\text{GUE}}^{(6)}$, where the numerator is the distance of the model to the analytic GUE average, and the denominator is the distance from the numerical GUE average to the analytical GUE average. Then for the largest system size we probed, $A, B = 3, 5$ spins, the ratio to the GUE was plotted against the ratio to

Poisson, for all the models discussed. It paints a picture of which models are best described by our methods. See figure 19.

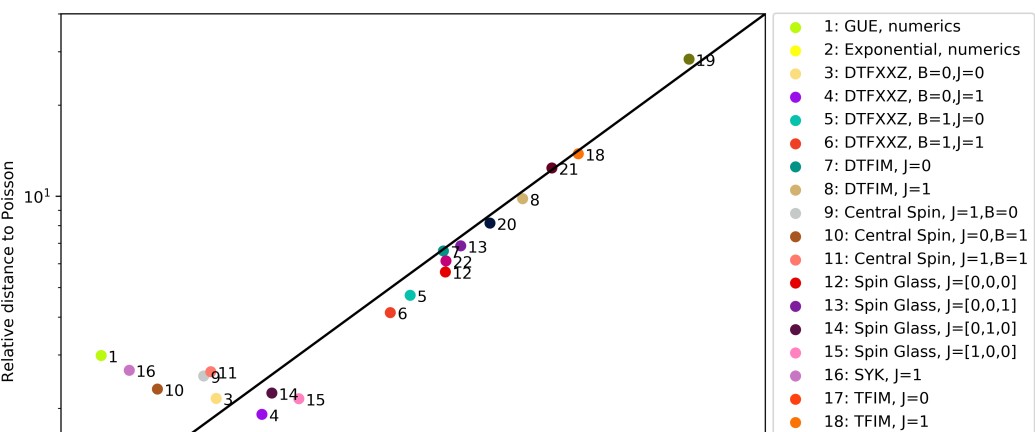

Figure 19: Relative distance of dynamics, in the sense of equation 51 of various models to the GUE and Poisson ensembles. Note that by our normalization the horizontal coordinate of the GUE ensemble must be unit, as well as the vertical of the Exponential. The closer another model approaches these points, the better its dynamics are approximated by the theory in this work.

We expect this behavior to continue into larger system sizes, and posit that the SYK model in general is most closely related in dynamics to the GUE, followed by the chaotic versions of the CS. Conversely, the XXZ in many incarnations doesn't equilibrate at all, like the TFIM, and is very dissimilar to both our analytical ensembles. Yet others land somewhere in between. For a visualisation of the aptitude of our methods for these models at other subsystem proportions than 3 coupled to 5 spins, see appendix E.

## 5.6 Conclusions and Outlook

In a nutshell, this work has shown how to average functions of the reduced density matrix over the GUE, as a general approach to understanding thermalization and entanglement generation. Along the way, we have gained understanding of calculation of general vertex operators and higher point correlators in the GUE field theory. Summarizing the experience in numerical works, we arrive at the following conclusion: information contained eigenvalue statistics are *not enough* to classify thermalization properties, we can show that ensembles may have same energy statistics as GUE, but don't thermalize in the same way. Notably, a model such as SYK does. A natural following step would be to focus on eigenfunction and matrix elements statistics using similar tools.

A fair critique of this research is that the averages are more of mathematical than physical significance. Most of the sets of eigenvectors $V \in \mathcal{U}(d)$ are highly non-local, exotic interactions. This criticism is treated by section 5.5. It appears many models do not allow themselves to be described by these results, but some, such as the SYK and CS models, do. Returning to the analytic matrix averages, it would be insightful to take averages over ensembles that

are non-uniform on $\mathcal{U}(d)$. Attempts have been made in this direction, using an additional weight $\propto \tilde{\lambda} \operatorname{Tr}\left([V,\Omega][V,\Omega]^{\dagger}\right)$, which rewards $V$ that are in some sense 'close' to a favored $\Omega \in \mathcal{U}(d)$. However, the constraints on probability distributions leave us with little power to weigh systems too undemocratically. This means in particular that as $d$ becomes large, the corrections induced by this weight compared to the flat average vanish. All the while, the calculation becomes dramatically enlarged. More thought on this matter is welcome. Also, besides exponential and Gaussian, other distributions of energy may be interesting if they can be physically motivated.

For anyone wishing to reproduce this work, or related questions, the required Python code will be made available with the publication. Most notably, the script to perform all the exact unitary integrals is free for examination. This includes a subroutine that produces symbolic Weingarten functions corresponding to any conjugacy class, at any dimension using the Murnaghan-Nakayama rule and hook-length formula, which may be useful in general [53].

There are several points we would like to mention at the end. First, to compute time evolution of the entanglement entropy a common strategy is to employ a *replica symmetry trick*, $S = -\operatorname{Tr}(\rho \log \rho) = -(\partial/\partial n)(\log \operatorname{Tr}(\rho^n))|_{n=1}$. In order to carry this out, as it requires an analytic continuation, we would need to *analytically* evaluate an average of an arbitrary power of the density matrix. This would require control of unitary integrals of non-integer polynomial degree. This may be within reach by means of a generating function technique. Similar perhaps to a matrix model of the Gross-Witten type [54,55]. In future work, it would be worthwhile to consider this and other techniques to perform unitary integrals. It is the authors' hope generating functions for Weingarten functions might allow us to push past polynomials of $V$ to more general forms.

# Acknowledgements

We are endebted to J-S Caux, the promotor of the first author, for his patience and flexibility allowing this child to be brought to term under his aegis. Besides this, we must mention fruitful discussion with O. Gamayun, of the same group, from whose mind specifically expressions (26) and (27) sprang, as if by divine inspiration. Finally, we are thankful for the input of Maris Ozols, to whom the first author went for advice early in this project. He made the original suggestion to us to consider the unitary integrals such as they stand in section 3. That this suggestion was in fact, in light of [28] not actually original, none of us were aware at the time.

**Funding Information** The authors gratefully acknowledge the support of European Research Council under ERC Advanced grant No 743032 (DC). Additionally, this work is part of the Delta-ITP consortium, a program of the Netherlands Organization for Scientific Research (NWO) that is funded by the Dutch Ministry of Education, Culture and Science (OCW).

# Appendix A Random Matrix Theory

In this appendix, we will expound upon Random Matrix Theory (RMT) and the techniques of performing matrix integrals. As stated in the setup, the idea is to model Hamiltonians as Hermitian random matrices. A Hermitian matrix ensemble can be sampled by choosing matrix elements independently at random, subject to the hermicity condition. The degrees of freedom are the real parts of the diagonal, as well as the real and imaginary parts of the superdiagonal elements.

However, in Quantum Mechanics we are not so much interested in the individual elements of $H$, but rather its eigenvectors and eigenvalues. We wish to change coordinates. This is analogous to converting a $2d$ Cartesian integral into a polar coordinates, hence the reference to angular and radial parts. It is conventional to split a uniform measure $dH = d(V\Lambda V^\dagger)$ over $(d \times d)$ Hermitian matrices $H \in \mathcal{M}$ into a product of the radial-angular form [56]:

$$dH = \prod_{j=1}^{d} dH_{jj} \prod_{k<l} d\text{Im}(H_{kl}) d\text{Re}(H_{kl}) = \mathcal{D}E \cdot dV \cdot \Delta^2(E). \tag{52}$$

In the middle expression, each factor is the familiar Lebesgue measure on the the independent variables of the matrix element. Then on the RHS, $\mathcal{D}E := \prod_{l=1}^{d} dE_l$ with each $dE_l$ the Lebesgue measure on an eigenvalue. $dV$ is the symbolic *Haar* measure on the unitary group. The Haar measure is the unique uniform, translationally invariant normalized measure on any compact Lie group, and by definition satisfies

$$\int_{\mathcal{U}(d)} dV = \int_{\mathcal{U}(d)} d(WV) = \int_{\mathcal{U}(d)} d(VW) = 1, \quad \forall\, W \in \mathcal{U}(d), \tag{53}$$

by the transitive nature of the group. In particular, using this measure, every eigenbasis $V$ is equally probable.

Finally, $\Delta(E)$ is the *Vandermonde determinant*. Its square is the Jacobian of the coordinate transformation $H \mapsto (V, E)$, defined in (8).

We see that for uniformly distributed elements of $H$, the eigenvalues are spaced apart: they repel. It is quite remarkable that such correlated eigenvalues result from uncorrelated simple elements. It can be unerstood by analogy. In polar coordinates the size of the orbit of the angle scales with the radius, which is reflected in the Jacobian. Analogously, the orbit in $\mathcal{M}$ of action of the unitary group through the eigenvectors of $H$ scales with $\Delta^2(E)$, and becomes measure zero when two eigenvalues are degenerate [40], and $dH$ vanishes.

If we consider a weight on the matrices, we desire it to be basis independent, just like the measure. One way to accomplish this is as follows:

$$\tilde{\mathcal{P}}(H) = \tilde{\mathcal{C}} \exp\left( -\sum_m \frac{\lambda_m}{m!} \text{Tr}(H^m) \right) = \tilde{\mathcal{C}} \exp\left( -\sum_m \frac{\lambda_m}{m!} \left( \sum_{l=1}^{d} E_l^m \right) \right). \tag{54}$$

Then if we collapse the constants to $\lambda_m = \lambda \delta_{m,2}$, we recover the *Gaussian Unitary Ensemble* (GUE). This weight is particularly useful, because it is easy to calculate averages with scalar Gaussian distributions, which is now essentially what we have on the energies. With suggestive notation, we see the weight only depends on the energies, $\mathcal{P}(H) = \mathcal{P}(E)$, leading to expression (7).

We demand for a well defined distribution

$$\int_{\mathcal{M}} \mathcal{P}(H) dH = \int_{\mathbb{R}^d} \mathcal{D}E \mathcal{P}(E) \Delta^2(E) \int_{\mathcal{U}(d)} dV = 1. \tag{55}$$

It is known in this ensemble that the normalization $\mathcal{C} = \mathcal{C}(d, \lambda)$ is fixed by [43]

$$\mathcal{C}^{-1} = \int_{\mathbb{R}^d} \mathcal{D}E \exp\left( -\frac{\lambda}{2} \sum_{l=1}^{d} E_l^2 \right) \Delta^2(E) = \sqrt{\frac{(2\pi)^d}{\lambda^{d^2}}} \prod_{j=1}^{d} j! \tag{56}$$

because $dV$ is normalized by definition[15].

---

[15]Technically, expression (52) and below should also incorporate the overcounting factor $(d!)^{-1}(2\pi)^{-d}$ [40] from the decomposition in (1), but this would simply be canceled again in the definition of $\mathcal{C}$.

In practice, we can obtain matrices from the GUE as follows. The real and imaginary parts of the superdiagonal elements can each be drawn independently from a $\mathcal{N}\left(0, \frac{1}{2}\lambda^{-1}\right)$ distribution, and the real diagonals from a $\mathcal{N}\left(0, \lambda^{-1}\right)$ distribution, and the matrix completed by the hermicity demand. The product of their independent distributions will result in a joint probability distribution proportional to $\exp(-\frac{\lambda}{2}\operatorname{Tr}\left[H^T H\right])$ as desired[16].

The stage is set: we have constructed a distribution and are still free to choose any function to average over it. To simplify equations in this work, we will fix $\lambda = 1$. We may do this because $\lambda$ merely rescales time. We can understand this in the following way. The functions whose expectation value we seek come from the Schrödinger equation and will always have time multiplying differences of energy. These yield calculations of the form

$$\langle\langle f \rangle\rangle := \int_{\mathcal{M}} dH \mathcal{P}(E) f\big(t(E_k - E_j)\big), \tag{57}$$

for $f$ some function in general depending on all $t(E_k - E_j)$, $j, k \in \{1, \ldots, d\}$. Rescaling the standard deviation of energies, $\sqrt{\lambda}$ for the GUE, is thus equivalent to dilating time. To restore the spread, one can substitute $t \mapsto t/\sqrt{\lambda}$ in the final result[17]. From this observation, the only truly 'free' parameters in this research of any consequence are $d_A$ and $d_B$.

## A.1 Nontrivial Unitary Integrals

Moving on to solution techniques, the integrals over energy (eigenvalues) can be performed in a standard way from complex analysis. By contrast, the integrals over the unitary group will require more modern machinery. Viewed as $d \times d$ matrices, there are standard identities for integrals of polyvariate monomials of matrix elements of $V \in \mathcal{U}(d)$ of degree $q$ and their Hermitian conjugates. They involve *Weingarten functions* [58] [59].

$$\int_{\mathcal{U}(d)} dV \, V_{i_1, j_1} \ldots V_{i_q, j_q} V^{\dagger}_{j'_1, i'_1} \ldots V^{\dagger}_{j'_q, i'_q} =$$
$$\sum_{\sigma, \tau \in S_q} \delta_{i_1, i'_{\sigma(1)}} \ldots \delta_{i_q, i'_{\sigma(q)}} \delta_{j_1, j'_{\tau(1)}} \ldots \delta_{j_q, j'_{\tau(q)}} \mathrm{Wg}(d, \sigma\tau^{-1}) = \tag{58}$$
$$\sum_{\sigma, \tau \in S_q} \delta_{I, \sigma(I')} \delta_{J, \tau(J')} \mathrm{Wg}(d, \sigma\tau^{-1}).$$

Here we have made the identification of the ordered set $I = \{i_1, \ldots, i_q\}$ and similar for $I', J, J'$, and the delta function $\delta_{I, \sigma(I')}$ is only satisfied when the full sets agree element-wise. From this equality, any polynomial integral is found by linearity.

We have also referenced the symmetric group of permutations of $q$ symbols, $S_q$. We will use notation that $S_q$ consists of all bijective maps $\{1, 2, \ldots q\} \to \{1, 2, \ldots q\}$. It is implied that any integral such as $\int V_{1,1} dV$ that does not contain an equal number of factors of $V$ and $V^{\dagger}$ will vanish by symmetry, but they will also not arise in this work.

The Weingarten functions $\mathrm{Wg}(d, \sigma)$, which can be calculated to any dimension using representation theory of $S_q$, in fact depend on the dimension $d$, and only on the conjugacy class of the permutation $\sigma$, not the specific element. As can be learned from any text on finite group theory, the conjugacy classes in $S_q$ are determined by cycle type, characterized by a *partition*

---

[16]Computationally, one can also generate two auxiliary real matrices $A_1, A_2$ of $(d \times d)$ i.i.d. $\mathcal{N}(0, \lambda^{-1})$ random variables, and construct $H = \frac{1}{2}\left((A_1 + A_1^T) + i(A_2 - A_2^T)\right)$ and it will have the desired statistical properties. [57]

[17]Rigorously, first substitute $t = \sqrt{\lambda}\tau$ in (57), and then absorb $\sqrt{\lambda}$ in the integrand by substituting $\sqrt{\lambda}E_j = x_j$. $\lambda$ will drop from the Gaussian, and the powers of $\lambda$ from the measure and Vandermonde determinant will cancel against those from the modified normalization constant $\mathcal{C}$. $\tau$ is now rescaled time.

$\phi = \{\phi_1, \ldots, \phi_l\}$ of the integer $q$, satisfying $\phi_1 \geq \phi_2 \geq \ldots \phi_l \geq 1$ and $\sum_{j=1}^{l} \phi_j = q$. Then we write $\phi(\sigma) \vdash q$. This means, for each $q$, there are finitely many Weingarten functions, which are quotients of polynomials in d. In this paper, we will only need the functions for $q = 2$ and $q = 4$. The denominator is $\prod_{z=0}^{q-1}(d^2 - z^2)$ for both cases[18]. The numerators can be found in table 1 below[19]. For example, taking $\sigma = (12)(34) \in S_4$, $\mathrm{Wg}(d, \sigma) = (d^2 + 6)/\big(d^2(d^2 - 1)(d^2 - 4)(d^2 - 9)\big)$.

Table 1: $\mathrm{Wg}(d, \sigma)$ functions

| $\phi(\sigma) \vdash q = 2$ | numerator $\mathrm{Wg}(d, \sigma)$ | $\phi(\sigma) \vdash q = 4$ | numerator $\mathrm{Wg}(d, \sigma)$ |
|---|---|---|---|
| $\{1, 1\}$ | $d^2$ | $\{1, 1, 1, 1\}$ | $d^4 - 8d^2 + 6$ |
| $\{2\}$ | $-d$ | $\{2, 1, 1\}$ | $-d^3 + 4d$ |
| | | $\{2, 2\}$ | $d^2 + 6$ |
| | | $\{3, 1\}$ | $2d^2 - 3$ |
| | | $\{4\}$ | $-5d$ |

## A.2 Orthogonal Polynomials and Symmetric Kernels

In section 4, it will be imperative to calculate the expectation value of functions that depend on energy over the distribution $\mathcal{P}(E)\Delta^2(E)\mathcal{D}E$ from the GUE. We will be interested in large $d$ behaviour, so straightforward evaluation of the $d$-dimensional energy integral is prohibitively complicated due to proliferation of factors in $\Delta(E)$. Luckily we will in practice be in need of $\langle f \rangle$ of functions $f(E_1, E_2, \ldots, E_n)$, with $n \leq d$, where $n$ does not scale with system size. Such an average is naturally evaluated in the basis of Orthogonal Polynomials (OPs). This is an indexed family of univariate polynomials $p_\mu(x)$, $\mu \in \{0, 1, \ldots\}$ together with a weight function $w(x)$. OPs obey a form of orthogonality according to the inner product

$$\big(p_\mu, p_\nu\big)_w := \int_{\mathbb{R}} dx\, p_\mu(x), p_\nu(x) w(x) = \big(p_\mu, p_\mu\big)_w \delta_{\mu,\nu}. \tag{59}$$

We cite from Forrester [43], due originally to Dyson [61]:

$$\frac{d!}{(d-n)!} \prod_{l=1}^{n} w(x_l) \frac{\int_{\mathbb{R}} dx_{n+1} w(x_{n+1}) \ldots \int_{\mathbb{R}} dx_d w(x_d) \Delta^2(x)}{\int_{\mathbb{R}} dx_1 w(x_1) \ldots \int_{\mathbb{R}} dx_d w(x_d) \Delta^2(x)} = \det_{1 \leq j, k \leq n} [K_d(x_j, x_k)] \tag{60}$$

for $n \leq d$, where the symmetric kernel $K_d(x, y)$ is given by

$$K_d(x, y) := \frac{\sqrt{w(x)w(y)}}{(p_{d-1}, p_{d-1})_w} \frac{p_d(x)p_{d-1}(y) - p_{d-1}(x)p_d(y)}{x - y} = \sqrt{w(x)w(y)} \sum_{\mu=0}^{d-1} \frac{p_\mu(x)p_\mu(y)}{(p_\mu, p_\mu)_w}. \tag{61}$$

The final equality is known as the *Christoffel-Darboux formula* [43]. This allows us to work with a normalized expression where all but $n$ dimensions have already been integrated out. It reduces the $d$ dimensional integral to an $n$ dimensional one, which will prove crucial. One more ingredient is needed for the determinant above. When taking the limit $y \to x$, expression (61) gives

$$K_d(x, x) = \frac{w(x)}{(p_{d-1}, p_{d-1})_w} \big(p'_d(x)p_{d-1}(x) - p'_{d-1}(x)p_d(x)\big). \tag{62}$$

---

[18]Caution: this structure does not persist exactly in higher $q$.

[19]These expressions are only correct for $d \geq 4$, luckily this is the smallest composite Hilbert space and they will not be applied to smaller $d$. See, e.g. [60].

In our case, the independent variables become $\{x_j\} \to \{E_j\}$ and the weights are Gaussian, $w(x_j) \to \exp\left(-\frac{1}{2}E_j^2\right)$. Indeed, then the denominator of (60) is just the normalization defined in (56). The OPs corresponding to the standard Gaussian distribution are the probabilist's Hermite polynomials, where we will keep the notation $\{p_\mu\}$ instead of the conventional $\{H_\mu\}$ to avoid confusion with the Hamiltonian. $p_\mu$ is a monic polynomial of degree $\mu$. Elementarily, these satisfy

$$(p_\mu, p_\nu)_w = \sqrt{2\pi}\mu!\delta_{\mu,\nu} \tag{63}$$

and

$$p'_\mu(x) = \mu p_{\mu-1}(x), \quad x p_\mu(x) = p_{\mu+1}(x) + p'_\mu(x). \tag{64}$$

A related set of polynomials are the generalized Laguerre polynomials, which have two indices.

$$L_\mu^{(\alpha)}(x) := \sum_{\nu=0}^{\mu} \binom{\mu+\alpha}{\mu-\nu}\frac{(-x)^\nu}{\nu!}. \tag{65}$$

They will become relevant through the useful identity [62]:

$$\int dx\, w(x) p_\mu(x+y) p_\nu(x+z) = \sqrt{2\pi}\mu! z^{\nu-\mu} L_\mu^{(\nu-\mu)}(-yz), \quad \mu \leq \nu. \tag{66}$$

Here $w(x)$ is still the standard Gaussian. Of the Laguerre polynomials, these identities will be needed [62]:

$$\begin{aligned}
x L_{\mu-1}^{(\alpha+1)}(x) &= (\mu+\alpha) L_{\mu-1}^{(\alpha)}(x) - \mu L_\mu^{(\alpha)}(x), \\
L_\mu^{(\alpha)}(x) &= L_\mu^{(\alpha+1)}(x) - L_{\mu-1}^{(\alpha+1)}(x).
\end{aligned} \tag{67}$$

$$L_\mu^{(\alpha+1)}(x) = \sum_{\nu=0}^{\mu} L_\nu^{(\alpha}(x). \tag{68}$$

## Appendix B  Unitary Technicalities of the Purity

In this appendix, we will work through the details of averaging the subsystem purity over the eigenbasis $V \in \mathcal{U}(d)$. Combining expressions (5) and (6), we find[20]

$$\begin{aligned}
\langle \gamma(t) \rangle &:= \int_{\mathcal{U}(d)} dV \,\mathrm{Tr}_A(\rho_A^2(t)) = \int_{\mathcal{U}(d)} dV \sum_{g_B,h_B=1}^{d_B} \sum_{g_A,h_A=1}^{d_A} \sum_{j,k,l,m=1}^{d} \\
&\quad \left( V_{(g_A,g_B)j} V_{1k} V_{j1}^\dagger V_{k(h_A,g_B)}^\dagger e^{i(E_k-E_j)t} V_{(h_A,h_B)l} V_{1m} V_{l1}^\dagger V_{m(g_A,h_B)}^\dagger e^{i(E_m-E_l)t} \right) = \\
&\quad \int_{\mathcal{U}(d)} dV\, V_{(g_A,g_B)j} V_{1k} V_{(h_A,h_B)l} V_{1m} V_{j1}^\dagger V_{k(h_A,g_B)}^\dagger V_{l1}^\dagger V_{m(g_A,h_B)}^\dagger e^{i(E_k+E_m-E_j-E_l)t}.
\end{aligned} \tag{69}$$

---

[20]Summation is made explicit in the third form of (69), afterwards it is omitted, but all indices are summed over in this entire appendix. Again, the subscript of the index indicates the range: e.g. $l \in \{1,\dots,d\}, g_B \in \{1,\dots,d_B\}, g_A \in \{1,\dots,d_A\}, j \in \{1,\dots,d\}$.

From this expression, we populate the multi-indices needed for expression (58).

$$\begin{aligned} I &= ((g_A, g_B), 1, (h_A, h_B), 1); & I' &= (1, (h_A, g_B), 1, (g_A, h_B)) \\ J &= (j, k, l, m); & J' &= (j, k, l, m). \end{aligned} \tag{70}$$

As before, the terms dependent of $I, I'$ decouple from those of $J, J'$. We can construct new $R_\sigma, Q_\tau$, which are similar to those used averaging the density matrix, although now they are indexed by permutations of $S_4$, and are thus $4! = 24$ dimensional. Also, for scalar purity, these are all scalar expressions. Then our final answer will take the form

$$\langle \gamma(t) \rangle = \sum_{\sigma, \tau \in S_4} R_\sigma Q_\tau \mathrm{Wg}(d, \sigma \tau^{-1}), \tag{71}$$

with

$$R_\sigma := \delta_{I, \sigma(I')}, \quad Q_\tau := \delta_{J, \tau(J')} e^{i(E_k + E_m - E_j - E_l)t}. \tag{72}$$

For the results, see table 2:

| Table 2: $\mathrm{Tr}(\rho_A^2(t))$ Sum Factors | | | | | |
|---|---|---|---|---|---|
| $\sigma, \tau \in S_4$ | $R_\sigma$ | $Q_\tau$ | $\sigma, \tau \in S_4$ | $R_\sigma$ | $Q_\tau$ |
| Id | $1$ | $\chi^2(t)$ | (12) | $d_B$ | $d \cdot \chi(t)$ |
| (123) | $d_B$ | $\chi(t)$ | (13) | $1$ | $\iota^2(t)\iota(-2t)$ |
| (132) | $d_A$ | $\chi(t)$ | (14) | $d_A$ | $d \cdot \chi(t)$ |
| (124) | $d_B$ | $\chi(t)$ | (23) | $d_A$ | $d \cdot \chi(t)$ |
| (142) | $d_A$ | $\chi(t)$ | (24) | $1$ | $\iota^2(-t)\iota(2t)$ |
| (134) | $d_B$ | $\chi(t)$ | (34) | $d_B$ | $d \cdot \chi(t)$ |
| (143) | $d_A$ | $\chi(t)$ | (1234) | $d \cdot d_B$ | $d$ |
| (234) | $d_B$ | $\chi(t)$ | (1243) | $d_B$ | $d$ |
| (243) | $d_A$ | $\chi(t)$ | (1324) | $d_A$ | $d$ |
| (12)(34) | $d \cdot d_B$ | $d^2$ | (1342) | $d_B$ | $d$ |
| (13)(24) | $1$ | $\chi(2t)$ | (1423) | $d_A$ | $d$ |
| (14)(23) | $d \cdot d_A$ | $d^2$ | (1432) | $d \cdot d_A$ | $d$ |

We will go through the derivation and definition of terms briefly. Starting with $R_\sigma$, a look at (70) tells us any permutation $\sigma$ taking the pairs $\{1, 3\} \to \{1, 3\} \subset \{1, 2, 3, 4\}$ forces all indices to 1, and the sum in (72) is trivially unit,

$$R_{\mathrm{Id}} = \delta_{g_A,1}^2 \delta_{g_B,1}^2 \delta_{h_A,1}^2 \delta_{h_B,1}^2 = 1 = R_{(13)} = R_{(24)} = R_{(13)(24)}. \tag{73}$$

Next,

$$R_{(12)} = \delta_{1,1} \delta_{(g_A, g_B),(h_A, g_B)} \delta_{(h_A, h_B),1} \delta_{1,(g_A, h_B)} = \delta_{g_B, g_B} \delta_{g_A, h_A} \delta_{h_A, 1} \delta_{h_B, 1}^2 \delta_{g_A, 1} = d_B, \tag{74}$$

as $g_B$ is left free. By the same token, any $\sigma$ that maps $1 \mapsto 2 \vee 3 \mapsto 4$ will leave a single free bath index.

$$R_{(12)} = R_{(34)} = R_{(123)} = R_{(124)} = R_{(134)} = R_{(234)} = R_{(1243)} = R_{(1342)} = d_B. \tag{75}$$

Conversely, for $\sigma : 1 \mapsto 4 \vee 3 \mapsto 1$, there is a single free subsystem index

$$R_{(14)} = R_{(23)} = R_{(132)} = R_{(142)} = R_{(143)} = R_{(243)} = R_{(1324)} = R_{(1423)} = d_A. \tag{76}$$

Lastly, mapping $1 \mapsto 4 \wedge 3 \mapsto 2$ leaves both subsystem indices undetermined, and equates the bath indices to each other, killing one summation

$$R_{(14)(23)} = \delta_{(g_A,g_B),(g_A,h_B)}\delta^2_{1,1}\delta_{(h_A,h_B),(h_A,g_B)} = \delta_{g_A,g_A}\delta^2_{g_B,h_B}\delta_{h_A,h_A} = d_A^2 d_B = d \cdot d_A = R_{(1432)}. \quad (77)$$

Likewise,

$$R_{(12)(34)} = \delta_{g_B,g_B}\delta^2_{g_A,h_A}\delta_{h_B,h_B} = d_A d_B^2 = d \cdot d_B = R_{(1234)}. \quad (78)$$

Moving on to $Q_\tau$, we note that $J = J'$, so $\delta_{J,\tau(J')} = \delta_{J,J'} = \delta_{j,j}\delta_{k,k}\delta_{l,l}\delta_{m,m}$, and the first element factorizes

$$Q_{\mathrm{Id}} = \delta_{J,J'}e^{i(E_k+E_m-E_j-E_l)t} = \left(\sum_{j,k=1}^d e^{i(E_k-E_j)t}\right)\left(\sum_{l,m=1}^d e^{i(E_m-E_l)t}\right) = \chi^2(t). \quad (79)$$

Where we are reminded of the definition of $\chi(t)$ in (15)

$$Q_{(12)} = \delta^2_{j,k}\delta_{l,l}\delta_{m,m}e^{i(E_k+E_m-E_j-E_l)t} = \sum_{j,l,m=1}^d e^{i(E_m-E_l)t} = d \cdot \chi(t), \quad (80)$$

with $\tau$ producing $\delta_{j,m} \vee \delta_{l,k} \vee \delta_{l,m}$, the same results:

$$Q_{(12)} = Q_{(14)} = Q_{(23)} = Q_{(34)} = d \cdot \chi(t). \quad (81)$$

Making use of definition (14), we have the pair

$$Q_{(13)} = \delta^2_{j,l}\delta_{k,k}\delta_{m,m}e^{i(E_k+E_m-E_j-E_l)t} = \iota^2(t)\iota(-2t) = \sum_{j,k,m=1}^d e^{i(E_m+E_k-2E_j)t},$$

$$Q_{(24)} = \delta^2_{k,m}\delta_{j,j}\delta_{l,l}e^{i(E_k+E_m-E_j-E_l)t} = \iota^2(-t)\iota(2t) \equiv \sum_{j,k,l=1}^d e^{i(2E_k-E_j-E_l)t}. \quad (82)$$

On to the first three-cycle, $\tau = (123)$. Here, as with all three-cycles, 3 out of 4 indices are equated, such as $j = k = l$, and the 4th, $h$ is left free. Always, two of the three equated indices kill their energies in the exponent, i.e. $E_k - E_j - E_l \mapsto -E_j$, and the 4th, $+E_m$ has opposite sign. All three-cycles contribute a factor:

$$Q_{(123)} = \sum_{j,k,l,m=1}^d \delta_{j,k}\delta_{k,l}\delta_{l,j}\delta_{m,m}e^{i(E_k+E_m-E_j-E_l)t} = \sum_{j,m=1}^d e^{i(E_m-E_j)t} \equiv \chi(t)$$

$$= Q_{(132)} = Q_{(124)} = Q_{(142)} = Q_{(134)} = Q_{(143)} = Q_{(234)} = Q_{(243)}. \quad (83)$$

Analogously, all four-cycles equate all indices $j = k = l = m$. Three summations kill four Kronecker-deltas, as the last is redundant (cyclic).

$$Q_{(1234)} = \delta_{j,k}\delta_{k,l}\delta_{l,m}\delta_{m,j}e^{i(E_k+E_m-E_j-E_l)t} = \sum_{j=1}^d e^0 = d$$

$$= Q_{(1243)} = Q_{(1324)} = Q_{(1342)} = Q_{(1423)} = Q_{(1432)}. \quad (84)$$

Continuing, in $\tau = (12)(34)$, two Kronecker delta's cancel the whole exponent, leaving two free summations:

$$Q_{(12)(34)} = \delta_{j,k}^2 \delta_{l,m}^2 e^{i(E_k + E_m - E_j - E_l)t} = \sum_{j,l=1}^{d} e^0 = d^2 = Q_{(14)(23)}. \tag{85}$$

And finally,

$$Q_{(13)(24)} = \delta_{j,l}^2 \delta_{k,m}^2 e^{i(E_k + E_m - E_j - E_l)t} = \sum_{j,k=1}^{d} e^{i(2E_k - 2E_j)t} = \chi(2t). \tag{86}$$

We notice that exchanging indices $2 \leftrightarrow 4$ in $\sigma$ has the effect of exchanging $A \leftrightarrow B$ in $R_\sigma$, or if the permutation is invariant, $R_\sigma = 1$. Conversely, this exchange in $\tau$ always leaves $Q_\tau$ invariant. This advises us that any multiple of $d_A$ in the final expression will also be present in $d_B$. The straightforward way to find the final expression, is by constructing the Weingarten matrix.

In the basis index order $\mathrm{Id}, (123), (132), \ldots, (1432)$ defined by table 4, the first few rows and columns of the Weingarten matrix look like

$$\mathrm{Wg}(d, \sigma\tau^{-1}) = \frac{1}{d^2(d^2-1)(d^2-4)(d^2-9)} \times$$
$$\begin{pmatrix}
d^4 - 8d^2 + 6 & 2d^2 - 3 & 2d^2 - 3 & 2d^2 - 3 & 2d^2 - 3 & \cdots \\
2d^2 - 3 & d^4 - 8d^2 + 6 & 2d^2 - 3 & 2d^2 - 3 & d^2 + 6 & \cdots \\
2d^2 - 3 & 2d^2 - 3 & d^4 - 8d^2 + 6 & d^2 + 6 & 2d^2 - 3 & \cdots \\
2d^2 - 3 & 2d^2 - 3 & d^2 + 6 & d^4 - 8d^2 + 6 & 2d^2 - 3 & \cdots \\
2d^2 - 3 & d^2 + 6 & 2d^2 - 3 & 2d^2 - 3 & d^4 - 8d^2 + 6 & \cdots \\
2d^2 - 3 & d^2 + 6 & 2d^2 - 3 & 2d^2 - 3 & d^2 + 6 & \cdots \\
\vdots & \vdots & \vdots & \vdots & \vdots & \ddots
\end{pmatrix}. \tag{87}$$

Performing the inner product was done symbolically by computer.

$$\sum_{\sigma,\tau \in S_4} R_\sigma Q_\tau \mathrm{Wg}(d, \sigma\tau^{-1}) = \frac{1}{d^2(d^2-1)(d^2-4)(d^2-9)} \times$$
$$\left\{ \left( \chi^2(t) + \chi(2t) + \iota^2(t)\iota(-2t) + \iota^2(-t)\iota(2t) - 4\chi(t) \right) \left[ d^4 - 2d^3 - 7d^2 + 8d + 12 \right] + \right.$$
$$(d_A + d_B) \left[ d^7 - d^6 - 13d^5 + 13d^4 + 36d^3 - 36d^2 - \right.$$
$$\left. \left( \chi^2(t) + \chi(2t) + \xi_0(t) + \xi_0^*(t) - 4\chi(t) \right) \left( d^3 - 3d^2 - 4d + 12 \right) \right] \right\} =$$
$$\frac{1}{d^2(d^2-1)(d^2-4)(d^2-9)} \times \left\{ \xi(t)(d+1)(d-2)(d+2)(d-3) + (d_A + d_B) \right.$$
$$\left. \cdot \left[ d^2(d-1)(d-2)(d+2)(d-3)(d+3) - \xi(t)(d-3)(d+2)(d-2) \right] \right\}. \tag{88}$$

In the last equality, we have defined another function $\xi(t)$ to hold the time and energy

dependence.

$$\xi(t) := \chi^2(t) + \chi(2t) + \iota^2(t)\iota(-2t) + \iota^2(-t)\iota(2t) - 4\chi(t)$$

$$= \left( d + 2\sum_{j<k} \cos\left((E_j - E_k)t\right)\right)^2 + 2\sum_{j,k,l} \cos\left((2E_j - E_k - E_l)t\right)$$

$$+ 2\sum_{j<k}\left( \cos\left(2(E_j - E_k)t\right) - 4\cos\left((E_j - E_k)t\right)\right) - 3d. \tag{89}$$

Another identity for $\xi(t)$, which is computationally favorable, is printed in (19).

At present the origin of the terms $(d \pm z)$, $z \in \mathbb{R}$ in expressions such as (88) is unclear to the authors, however their appearance is fortuitous: they cancel neatly with the denominator. The final result is then equation (18).

### B.1 Third power of the Density Matrix

Using similar techniques, any power of the density matrix is obtainable in principle. However, already for the third power, $(6!)^2 = 518400$ terms must be calculated and added. To this end, a Python routine was created that constructs $R_\sigma$ and $Q_\tau$, and performs the inner product. For a copy of this routine, please contact the first author. The result is printed below without derivation.

Recalling $\iota(t)$ from (14),

$$\zeta(t) := \left| \iota^3(t) + 2\iota(3t) + 3\iota(t)\iota(2t)\right|^2 - 36|\iota(t)|^2. \tag{90}$$

Using this, the trace of the third power of the reduced density matrix averages to

$$\int_{\mathcal{U}(d)} dV \operatorname{Tr}_A(\rho_A^3(t)) = \frac{(d_A + d_B)^2 + d + 1}{(d+1)(d+2)} + \frac{d + 1 - d_A - d_B}{d^2(d+1)^2(d+5)} \times$$

$$\left( \frac{(\zeta(t) - 9\xi(t))(1 - d_A - d_B)}{(d-1)(d+2)} + (d_A + d_B)\left( \frac{3\xi(t)}{d+3} + \frac{\zeta(t)}{(d-1)(d+4)}\right)\right). \tag{91}$$

## Appendix C  Derivation of n-Point Correlation Functions

In this appendix, we elaborate on the details of deriving expression (22) and what follows. To begin, we investigate the emergence of the generalized Laguerre polynomial $L_{d-1}^{(1)}(t^2) = \operatorname{Tr} F(t)$; $F(t)$ is defined in (23). An important component is the Fourier transform of the (diagonal element of the) symmetric kernel in (62).

$$\int_{\mathbb{R}} dE_1 K_d(E_1, E_1)e^{iE_1 t} =$$

$$\int_{\mathbb{R}} dE_1 \frac{e^{-\frac{1}{2}E_1^2 + iE_1 t}}{(p_{d-1}, p_{d-1})_w}\left( p_d'(E_1)p_{d-1}(E_1) - p_{d-1}'(E_1)p_d(E_1)\right) =$$

$$\frac{e^{-t^2/2}}{\sqrt{2\pi}(d-1)!}\int_{\mathbb{R}} d\tilde{E}_1 e^{-\frac{1}{2}\tilde{E}_1^2}\left( p_d'(\tilde{E}_1 + it)p_{d-1}(\tilde{E}_1 + it) - p_{d-1}'(\tilde{E}_1 + it)p_d(\tilde{E}_1 + it)\right). \tag{92}$$

Here we have completed the squares in the exponent

$$-\frac{1}{2}E_1^2 + iE_1 t = -\frac{1}{2}(E_1 - it)^2 + \frac{(it)^2}{2}, \tag{93}$$

and made the substitution of integration variable $\tilde{E}_1 := E_1 - it$. Then technically the integration domain is shifted up by a distance $t$ into the complex plane, to a line parallel to the real line. But we observe that the integrand dies at infinity and moreover has no poles. Then the domain can be deformed continuously back to its original position. Employing the first of expression (64) and subsequently (66) which expresses an integral of shifted Hermite polynomials in terms of generalized Laguerre polynomials, we arrive at

$$\int_{\mathbb{R}} dE_1 K_d(E_1, E_1) e^{iE_1 t} = e^{-\frac{1}{2}t^2}\left(d \cdot L_{d-1}^{(0)}(t^2) + t^2 L_{d-2}^{(2)}(t^2)\right) = e^{-\frac{1}{2}t^2} \cdot L_{d-1}^{(1)}(t^2). \tag{94}$$

In the last equality we made use of the two standard identities (67). This expression is manifestly a function of $(t^2)$, thus the sign of time doesn't matter. We continue by observing

$$\int_{\mathbb{R}} dE_1 K_d(E_1, E_1) e^{iE_1 t} \int_{\mathbb{R}} dE_2 K_d(E_2, E_2) e^{-iE_2 t} = e^{-t^2}\left(L_{d-1}^{(1)}(t^2)\right)^2. \tag{95}$$

This calculation will also prove useful in the larger correlators needed for the average purity. But first we move on to the origin of $F(t)$. Consider completing the square again and using the same substitution of $\tilde{E}_{1\frac{1}{2}\pm\frac{1}{2}} = E_{1\frac{1}{2}\pm\frac{1}{2}} \pm it$ in

$$
\begin{aligned}
&\int_{\mathbb{R}^2} dE_1 dE_2 K_d^2(E_1, E_2) e^{i(E_1 - E_2)t} \\
&= \int_{\mathbb{R}^2} dE_1 dE_2 w(E_1) w(E_2)\left(\sum_{\mu=0}^{d-1} \frac{p_\mu(E_1) p_\mu(E_2)}{(p_\mu, p_\mu)_w}\right)^2 e^{i(E_1 - E_2)t} \\
&= \left(e^{-t^2/2}\right)^2 \int_{\mathbb{R}^2} d\tilde{E}_1 d\tilde{E}_2 w(\tilde{E}_1) w(\tilde{E}_2)\left(\sum_{\mu=0}^{d-1} \frac{p_\mu(\tilde{E}_1 + it) p_\mu(\tilde{E}_2 - it)}{\sqrt{2\pi}\mu!}\right)^2 \\
&= \sum_{\mu,\nu=0}^{d-1} \frac{e^{-t^2}}{2\pi\mu!\nu!} \int_{\mathbb{R}^2} d\tilde{E}_1 d\tilde{E}_2 w(\tilde{E}_1) p_\mu(\tilde{E}_1 + it) p_\nu(\tilde{E}_1 + it) w(\tilde{E}_2) p_\mu(\tilde{E}_2 - it) p_\nu(\tilde{E}_2 - it).
\end{aligned}
\tag{96}
$$

We have also substituted the first equality of (61). From here, we construct

$$
\begin{aligned}
F_{\mu,\nu}(t) &:= \frac{e^{-\frac{1}{2}t^2}}{\sqrt{2\pi\mu!\nu!}} \int_{\mathbb{R}} d\tilde{E}_1 w(\tilde{E}_1) p_\mu(\tilde{E}_1 + it) p_\nu(\tilde{E}_1 + it) \\
&= e^{-\frac{1}{2}t^2} \frac{(\min\mu,\nu)!}{\sqrt{2\pi\mu!\nu!}} \sqrt{2\pi}(it)^{|\nu-\mu|} L_{\min\mu,\nu}^{(|\nu-\mu|)}(t^2),
\end{aligned}
\tag{97}
$$

by again invoking (66), leading immediately to definition (23). From there,

$$\int_{\mathbb{R}^2} dE_1 dE_2 K_d^2(E_1, E_2) e^{i(E_1 - E_2)t} = \sum_{\mu,\nu=0}^{d-1} F_{\mu,\nu}(t) F_{\mu,\nu}(-t) = \mathrm{Tr}[F(t)F(-t)], \tag{98}$$

and we have a full derivation of expression (22). Incidentally, this is the squared Frobenius norm of $F(t)$. Let us examine the matrix function. $F_{\mu,\nu}(t) \in \mathbb{R}$ for $\mu + \nu$ even, and $F_{\mu,\nu}(t) \in \mathbb{I}$ for $\mu + \nu$ odd. However its trace and traces of products of $F(t)$ will turn out to be all real.

$$F_{\mu,\nu}(t) = e^{-\frac{1}{2}t^2} \begin{pmatrix} 1 & it & \frac{-t^2}{\sqrt{2}} & i\frac{-t^3}{\sqrt{6}} & \cdots \\ it & -t^2+1 & i\frac{-t^3+2t}{\sqrt{2}} & \frac{-3t^2+t^4}{\sqrt{6}} & \cdots \\ \frac{-t^2}{\sqrt{2}} & i\frac{-t^3+2t}{\sqrt{2}} & \frac{1}{2}t^4-2t^2+1 & i\frac{t^5-6t^3+6t}{2\sqrt{3}} & \cdots \\ i\frac{-t^3}{\sqrt{6}} & \frac{-3t^2+t^4}{\sqrt{6}} & i\frac{t^5-6t^3+6t}{2\sqrt{3}} & -\frac{1}{6}t^6+\frac{3}{2}t^4-3t^2+1 & \cdots \\ \vdots & \vdots & \vdots & \vdots & \ddots \end{pmatrix}. \tag{99}$$

Miraculously, this definition of $F(t)$ allows us to reconcile the diagonal elements of the symmetric kernel with the off-diagonal, seeing as immediately from (97) and (68), $\operatorname{Tr}F(t) = \sum_\mu F_{\mu,\mu}(t) = e^{-\frac{1}{2}t^2}L_{d-1}^{(1)}(t^2)$. Moving on, $F(t)$ is defined in such a way that

$$F_{\mu,\nu}(0) = \delta_{\mu,\nu}. \tag{100}$$

This allows us to quickly check the normalization. For $t = 0$, a look at expressions (22) and (60) tells us that the integral in the numerator should equal that in the denominator. Indeed, setting $t = 0$ there gives

$$d(d-1)\langle e^0 \rangle = e^0 \left( \left( \sum_{\mu=0}^{d-1} \delta_{\mu,\mu} \right)^2 - \sum_{\mu,\nu=0}^{d-1} \delta_{\mu,\nu}^2 \right) = d^2 - d, \tag{101}$$

as desired.

## C.1 Three- and Four-Point Functions

In this subsection, we will continue the work above, and explain the derivation of the correlators that comprise the expectation value $\langle \xi(t) \rangle$.

Before we start though, with specific integrals, we will emphasize the pattern distilled from the previous calculation. The determinant of kernels in (60), by the Leibniz expansion, is a sum of products. Each product has factors of the form $K_d(E_j, E_k)$, which can be seen to 'couple' $E_j$ and $E_k$. There are also factors of the form $e^{ic_l E_l t}$, for $c_l \in \{\pm 1, \pm 2\}$ coming from the integrand. Each energy appears exactly twice in a kernel and once in an exponent. If $j = k$, energy $E_j$ is coupled to itself, and after integration results in a factor

$$\int dE_j K_d(E_j, E_j) e^{ic_j E_j t} = e^{-(c_j t)^2/2} L_{d-1}^{(1)}\left((c_j t)^2\right) = \operatorname{Tr} F(c_j t). \tag{102}$$

If $j \neq k$, then energies $E_j$ and $E_k$ are coupled to each other, resulting in factors

$$\int \dots K_d(E_j, E_k) e^{ic_j E_j t} e^{ic_k E_k t} \mapsto \sum_{\alpha=0}^{d-1} F_{\mu,\alpha}(c_j t) \cdot F_{\alpha,\nu}(c_k t) \dots \tag{103}$$

We again view these $F_{\mu,\nu}(c_j t)$ factors as symmetric $(d \times d)$ matrices[21], $\mu, \nu \in \{0, 1, \dots, d-1\}$, and the coupling as matrix multiplication. The string of coupled matrices closes on itself: it

---

[21]For a correlator in $d$ dimensions, we only use the upper left $d \times d$ subblock of the in principle infinite matrix. As we increase dimension, we can simply calculate more rows and columns.

forms a loop. This is accounted for by tracing over the matrix product. So dropping the indices, for instance

$$
\begin{aligned}
\int dE_1 dE_2 dE_3 K_d(E_1, E_2) K_d(E_2, E_3) K_d(E_3, E_1) e^{i(c_1 E_1 + c_2 E_2 + c_3 E_3)t} = \\
e^{-(c_1^2 + c_2^3 + c_3^2)\frac{t^2}{2}} \operatorname{Tr}\big[ F(c_1 t) F(c_2 t) F(c_3 t) \big],
\end{aligned}
\tag{104}
$$

and the generalization to larger loops is evident. Trivially, from (22)

$$
d(d-1)\big\langle e^{i(E_1 - E_2)(2t)} \big\rangle = \Big( \operatorname{Tr} F(2t) \cdot \operatorname{Tr} F(-2t) - \operatorname{Tr}[F(2t) F(-2t)] \Big).
\tag{105}
$$

For higher correlators, all that is left to do is to expand the determinant and collect like terms[22]. The three-point correlator was included in the main text above, in expression (29).

For the four-point correlator, at times the order of the non-commutative coupling will be important.

$$
\begin{aligned}
\frac{d!}{(d-4)!}\big\langle e^{i(E_1 + E_2 - E_3 - E_4)t} \big\rangle &= \int_{\mathbb{R}^3} dE_1 dE_2 dE_3 dE_4 \det_{1 \le j,k \le 4}[K_d(E_j, E_k)]) e^{i(E_1 + E_2 - E_3 - E_4)t} \\
&= \Big( \operatorname{Tr} F(t) \Big)^4 - 2\Big( \operatorname{Tr} F(t) \Big)^2 \operatorname{Tr} F^2(t) - 4\Big( \operatorname{Tr} F(t) \Big)^2 \operatorname{Tr}\big[ F(t) F(-t) \big] \\
&\quad + 8 \operatorname{Tr} F(t) \operatorname{Tr}\big[ F^2(t) F(-t) \big] + \Big( \operatorname{Tr} F^2(t) \Big)^2 + 2\Big( \operatorname{Tr}\big[ F(t) F(-t) \big] \Big)^2 \\
&\quad - 4 \operatorname{Tr}\big[ F(t) F(t) F(-t) F(-t) \big] - 2 \operatorname{Tr}\big[ F(t) F(-t) F(t) F(-t) \big].
\end{aligned}
\tag{106}
$$

The substitution can be made that $F(-ct) = I_\pm F(ct) I_\pm$ for $(I_\pm)_{0 \le \mu, \nu \le d-1} = (-1)^\mu \delta_{\mu,\nu}$, so $I_\pm^2 = \mathbb{1}$. This means, inside any trace of matrix products, we may substitute $t \to -t$, e.g. $\operatorname{Tr} F^2(-t) = \operatorname{Tr}\big[ I_\pm F(t) I_\pm I_\pm F(t) I_\pm \big] = \operatorname{Tr} F^2(t)$, allowing us to group terms. Incidentally, this is another manifestation that all correlators are real.

In general, this procedure is described by equation (28).

Combining equations (25), (22), (105), (29), and (106), we have all the information needed to find expression (25).

## C.2 Derivation of the Generating Function

In this subsection, we will prove that the generating function $G(\{a_m\})$ in equation (26) indeed procedurally generates equation (28). In the latter, let us say for concreteness we are looking for an $n$-point correlator characterized by the sequence $\{c_1, c_2, \dots c_n\}$, which in turn consists of $l < n$ distinct elements $\{m_k\}$ with respective multiplicities $\{n_k\}$. I.e. $\sum_{k=1}^l n_k = n$.

This is done using the Leibniz formula, notationally making use of the totally antisymmetric Levi-Civita tensor $\epsilon$.

$$
G(\{a_m\}) := \epsilon_{\mu_0, \dots, \mu_{d-1}} \prod_{\nu=0}^{d-1} \bigg( \delta_{\nu, \mu_\nu} + \sum_{m \in \mathbb{Z}} a_m F_{\nu, \mu_\nu}(mt) \bigg).
\tag{107}
$$

Anticipating the final result, we will focus our attention on the terms in this large product that have exactly a prefactor $\prod_j a_{c_j} = \prod_k \big( a_{m_k} \big)^{n_k}$. For simplicity, first assume all $c_j$ distinct, or all $n_k = 1$. We will later relax this assumption.

---

[22] Besides duplicates in the expansion of the determinant of a symmetric matrix, if in expression (29) we can obtain one term from another by exchanging $E_2 \leftrightarrow E_3$, they will result in the same integral.

As a combinatoric exercise, we can imagine drawing the term containing $a_{c_j}$ in the infinite series from any of the many factors $(\delta + \sum_m a_m \ldots)$, i.e. corresponding to any $\nu$. Once we have drawn one term from every factor, their product is one final term in the expanded product. Let us define $\nu_j$ to be the index of the factor where we select $a_{c_j}$. Then for any $n$ distinct values $\{\nu_1, \ldots, \nu_n\} \in \{0, \ldots, d-1\}^{\otimes n}$, we will obtain one term with the correct prefactors. From the excluded factors, all other values $\nu$, we multiply the delta $\delta_{\nu, \mu_\nu}$. The sum of any and all such terms out of (107) is then

$$G(\{a_m\}) = \sum_{\{\nu_1, \ldots, \nu_n\} \neq} \epsilon_{\mu_0, \ldots, \mu_{d-1}} \prod_{j=1}^{n} a_{c_j} F_{\nu_j, \mu_{\nu_j}}(c_j t) \prod_{\nu \notin \{\nu_1, \ldots, \nu_n\}} \delta_{\nu, \mu_\nu} + \mathcal{O}(\neq \prod_j a_{c_j}). \tag{108}$$

This second product of delta functions, after contraction, has the effect of setting the $\nu^{\text{th}}$ index of the Levi-Civita tensor to $\nu$. For example, if $n = 2, d = 5, \nu_1 = 0, \nu_2 = 3$, the corresponding term would be $\epsilon_{\mu_0, 1, 2, \mu_3, 4} a_{c_1} F_{0, \mu_0}(c_1 t) a_{c_2} F_{3, \mu_3}(c_2 t)$. Aside from the dummy variables $a_{c_j}$, this is simply the antisymmetrization over the remaining free indices, reducing the Levi-Civita symbol to $n$ effective dimensions, or the term to an $n \times n$ determinant

$$G(\{a_m\}) = \sum_{\{\nu_1, \ldots, \nu_n\} \neq} \prod_{j=1}^{n} a_{c_j} \det_{1 \leq j, k \leq n} F(c_j t)_{\nu_j, \nu_k} + \mathcal{O}(\neq \prod_j a_{c_j}). \tag{109}$$

We can now generalize to degenerate $c_j$: if some $c_j = c_{j'}, j \neq j'$, then in fact some of the terms in (108) correspond to the same combination in (107), and we must compensate for overcounting by dividing by $\prod_{k=1}^{l} n_k!$. And finally, any determinant of a matrix with duplicate columns vanishes, so we may promote the sum to include also sets of nondistinct $\{\nu_j\}$: these terms vanish regardless. Then we have arrived at the most general

$$G(\{a_m\}) = \sum_{\{\nu_1, \ldots, \nu_n\}=0}^{d-1} \prod_{k=1}^{l} \frac{(a_{m_k})^{n_k}}{n_k!} \det_{1 \leq j, k \leq n} F(c_j t)_{\nu_j, \nu_k} + \mathcal{O}(\neq \prod_j a_{c_j}), \tag{110}$$

which agrees with expression (27).

## Appendix D   Poisson Statistics Comparison

To contrast GUE-, also called Wigner-Dyson statistics of eigenvalues, there are Poisson statistics. Either is characterized in terms of the distribution of gaps between consecutive energy levels. For the former, as we have seen, levels repel due to the Vandermonde determinant and so the probability to find any $E_j$ approach $E_j + 1$ vanishes. The exact distribution of the gap $|E_1 - E_2|$ is not known[23] to arbitrary $d > 2$. By contrast in Poisson statistics, we don't necessarily know the distribution of single energies, but the gaps are weighted by their size according to the exponential distribution:

$$P(|E_j - E_{j+1}|) = \mu e^{-\mu |E_j - E_{j+1}|}. \tag{111}$$

The distribution is defined on $\mathbb{R}_{>0}$ and $\mu > 0$ is a free parameter. We are curious how the behavior of $\langle \chi(t) \rangle_P$ compares to that of $\langle \chi(t) \rangle_{\text{GUE}}$, averaged under different distributions of energy gap. Luckily $\chi(t)$ only depends on differences of energy. An exponential distribution of

---

[23]For $d = 2$ the exact result of the gap distribution is known as the *Wigner Surmise*, which is a good approximation to general dimension [63]. However, we work in arbitrary $d$ and do not require here that $E_1$ and $E_2$ are ordered and therefore adjacent.

gaps emerges if the probability to encounter a level is constant over $\mathbb{R}$ and does not depend on the levels before or after it: all levels are uncorrelated. Therefore we posit that this exponential behavior also persists between any two levels, not just adjacent ones, albeit with a different $\mu$. Then all terms in $\chi(t)$ have the same average, and analogous to the previous calculations, we simplify

$$\langle \chi(t) \rangle_P = d + d(d-1) \langle e^{i(E_1 - E_2)t} \rangle_P. \tag{112}$$

This gap distribution can be achieved by taking a product form joint PDF of exponential distributions on each separate energy.

$$
\begin{aligned}
\langle e^{i(E_1 - E_2)t} \rangle_P &= \int_0^\infty dE \mu^d \prod_j e^{-\mu E_j} e^{i(E_1 - E_2)t} \\
&= \mu^2 \int_0^\infty dE_1 \int_0^\infty dE_2 e^{E_1(it - \mu) - E_2(it + \mu)} = \frac{\mu^2}{\mu^2 + t^2},
\end{aligned}
\tag{113}
$$

as $\mu > 0$ by construction.

In order to interpret this result fairly, we must choose a value for $\mu$. Of the exponential distribution, it is known that average is $1/\mu$ and the variance is $1/\mu^2$. These readily yield

$$\langle (E_1 - E_2)^2 \rangle_P = \int_0^\infty (E_1 - E_2)^2 P(|E_1 - E_2|) d|E_1 - E_2| = \frac{2}{\mu^2}. \tag{114}$$

This is useful, because we will set the parameter $\mu = \mu(d)$ such that this second moment of the gaps agrees between the exponential distribution and the GUE, where the statistics also depend on $d$. Ideally, we would equate the first moment, but $\langle E_1 - E_2 \rangle_{\text{GUE}} = 0$ trivially and $\langle |E_1 - E_2| \rangle_{\text{GUE}}$ is not known.

A straightforward correlator calculation tells us,

$$
\begin{aligned}
d(d-1) \langle (E_1 - E_2)^2 \rangle_{\text{GUE}} &= \int_{\mathbb{R}} dE_1 \int_{\mathbb{R}} dE_2 \det_{1 \le j,k \le 2} K_d(E_j, E_k)(E_1 - E_2)^2 = \\
\int_{\mathbb{R}} dE_1 \int_{\mathbb{R}} dE_2 & \left( K_d(E_1, E_1) K_d(E_2, E_2)(E_1^2 + E_2^2 - 2E_1 E_2) - K_d^2(E_1, E_2)(E_1 - E_2)^2 \right).
\end{aligned}
\tag{115}
$$

We know $\int dx K_d(x,x) = d$ and it is also clear by symmetry that $\int dx K_d(x,x)x = 0$, by comparison to $d \cdot \langle E_1 \rangle_{\text{GUE}}$. Using these observations and substituting an alternative form of the kernel [43],

$$K_d(x, y) = \frac{\sqrt{w(x)w(y)}}{(p_{d-1}, p_{d-1})_w} \frac{p_d(x)p_{d-1}(y) - p_{d-1}(x)p_d(y)}{x - y}, \tag{116}$$

we transform to

$$
\begin{aligned}
d(d-1) \langle (E_1 - E_2)^2 \rangle_{\text{GUE}} &= d \int_{\mathbb{R}} dE_1 K_d(E_1, E_1) E_1^2 + d \int_{\mathbb{R}} dE_2 K_d(E_2, E_2) E_2^2 \\
- \int_{\mathbb{R}} dE_1 \int_{\mathbb{R}} dE_2 & \frac{w(E_1)w(E_2)}{2\pi(d-1)!^2} \left( p_d(E_1)p_{d-1}(E_2) - p_{d-1}(E_1)p_d(E_2) \right)^2.
\end{aligned}
\tag{117}
$$

In the last term, due to the orthogonal polynomials, the cross terms of the square vanish. We are left to use the inner product in expression (63):

$$\int_{\mathbb{R}} dE_1 \int_{\mathbb{R}} dE_2 \frac{w(E_1)w(E_2)}{2\pi(d-1)!^2} \left( p_d^2(E_1)p_{d-1}^2(E_2) + p_{d-1}^2(E_1)p_d^2(E_2) \right) =$$
$$\frac{2\pi d!(d-1)! + 2\pi(d-1)!d!}{2\pi(d-1)!^2} = 2d. \tag{118}$$

Furthermore, from (62), we use properties of Hermite polynomials in (64) repeatedly to modify

$$\int_{\mathbb{R}} dx K_d(x,x)x^2 = \int_{\mathbb{R}} dx \frac{w(x)}{\sqrt{2\pi}(d-1)!} \Bigg( d\big(p_d(x) + (d-1)p_{d-2}(x)\big)^2$$
$$- (d-1)\big(p_{d-1}(x) + (d-2)p_{d-3}(x)\big)\big(p_{d+1}(x) + d p_{d-1}(x)\big) \Bigg) \tag{119}$$
$$= \frac{\sqrt{2\pi}}{\sqrt{2\pi}(d-1)!} \Bigg( d\big(d! + (d-1)^2(d-2)!\big) - (d-1)d(d-1)! \Bigg) = d^2.$$

In the penultimate equality, we used orthogonality of the polynomials to discard most of the terms, and integrate the rest. Putting together (117), (118) and (119), we find

$$\langle (E_1 - E_2)^2 \rangle_{\text{GUE}} = \frac{1}{d(d-1)}(2 \cdot d \cdot d^2 - 2d) = 2(d+1). \tag{120}$$

We scale the Poisson statistics to agree on this value[24], $2/\mu^2 = 2(d+1) \Leftrightarrow \mu = (d+1)^{-1/2}$, so finally the Poisson average $\chi(t)$ is given by equation (39).

For sufficiently idealized ensembles of integrable systems, meaning specifically the model's eigenbases are uniformly Haar distributed over the Unitary group, we expect the subsystem dynamics to follow this time-evolution.

### D.1 Purity with Poisson Statistics

The same PDF used in expression (113) allows us to derive $\langle \xi(t) \rangle_P$, as a modification of (25). The result is

$$\langle \xi(t) \rangle_P = 4d(d-1)\frac{\mu^2}{\mu^2 + 4t^2} + \frac{4d!}{(d-3)!}\frac{\mu^4(\mu^2 + 3t^2)}{(\mu^2 + t^2)^2(\mu^2 + 4t^2)} + \frac{d!}{(d-4)!}\left(\frac{\mu^2}{\mu^2 + t^2}\right)^2$$
$$+ 4d(d-1)^2 \frac{\mu^2}{\mu^2 + t^2} + 2d(d-1). \tag{121}$$

This can be used to describe the time-dependent purity of ensembles of integrable systems.

## Appendix E   Numerical Models Comparisons

In this appendix, we will include as a reference a 'heatmap' of the relative distances of dynamics of the various discussed established models, for various subsystem sizes, to our analytical

---

[24]In general, with the GUE $\lambda \neq 1$, this becomes $\mu = \sqrt{\lambda/(d+1)}$.

ensembles. The distance is as defined in equation 51, and is subsequently normalized by the distance of the numerical implementation of either the GUE ensemble to its analytic counterpart, or the same distance for the Poisson. This normalization helps compare systems at different sizes. Without it, we are also measuring the dimensionality of the Hilbert space with a norm like 51 as much as we are measuring the dynamics. Furthermore it accounts for the numerical noise of sampling only a finite number of realizations from an ensemble, which might be more or less valid at different system sizes as well.

To illustrate, first we will display the relative distances of the numerical implementations of the GUE and Exponential ensembles themselves. Both have one map normalized to unity by default. See figures 20 and 21.

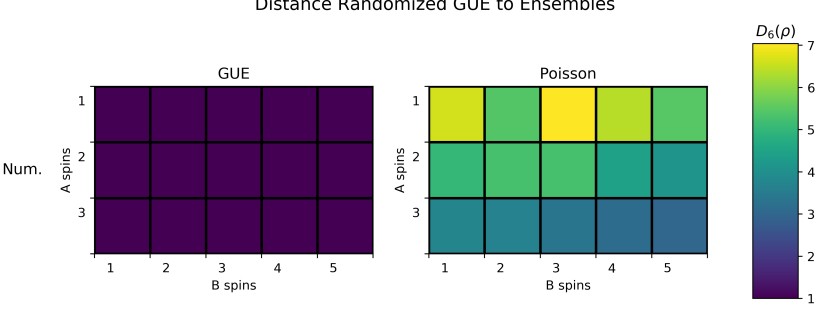

Figure 20: Left: Distance of subsystem dynamics of the numerical GUE to the analytical GUE, normalized to itself, Right: Distance of numerical GUE to the analytical Poisson distribution, normalized to the distance of the numerical Exponential Distribution to the analytical Poisson. These values are given for different number of subsystem and bath spins.

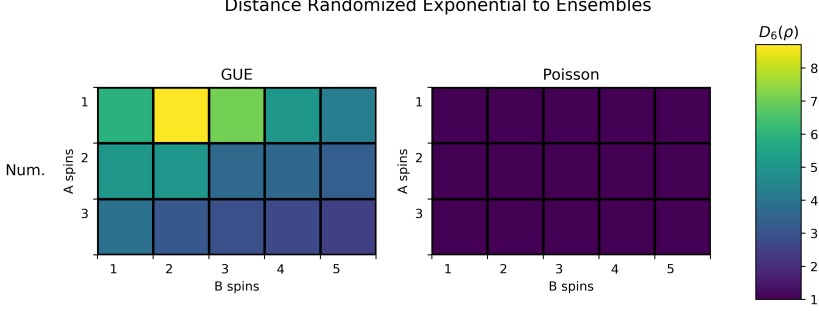

Figure 21: Left: Distance of subsystem dynamics of the numerical Poisson to the analytical GUE, normalized to the distance of the numerical GUE to the analytical GUE. Right: Distance of numerical Poisson to the analytical Poisson, normalized to itself. These values are given for different number of subsystem and bath spins.

Note that although the legend indicates this is the distance $D^{(6)}$ proper, it is in fact normalized, and note also that in the following figures, the legend scale varies from system to system quite strongly.

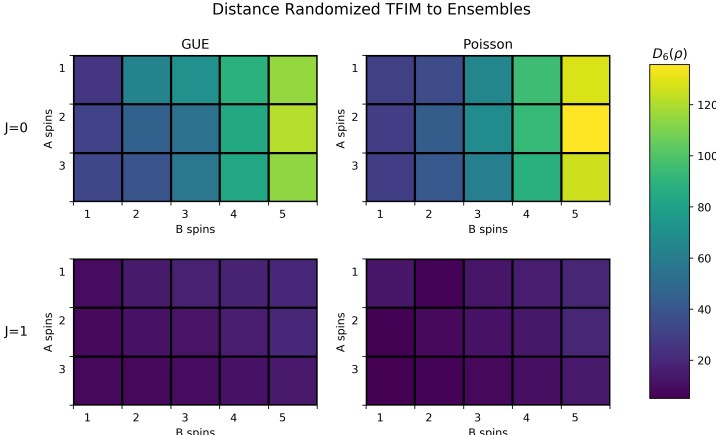

Figure 22: Left: Normalized distance of dynamics of Transverse Field Ising Model (see equation 42) to analytical GUE. Right: to analytical Poisson.

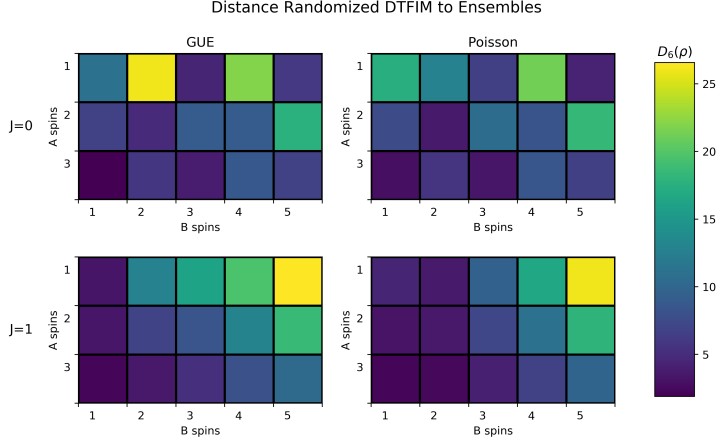

Figure 23: Left: Normalized distance of dynamics of Disordered Transverse Field Ising Model (see equation 43) to analytical GUE. Right: to analytical Poisson.

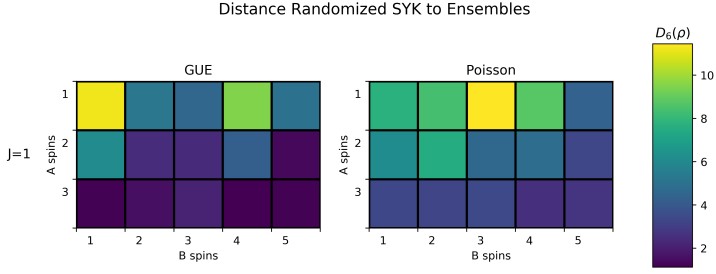

Figure 24: Left: Normalized distance of dynamics of SYK Model (see equation 46) to analytical GUE. Right: to analytical Poisson.

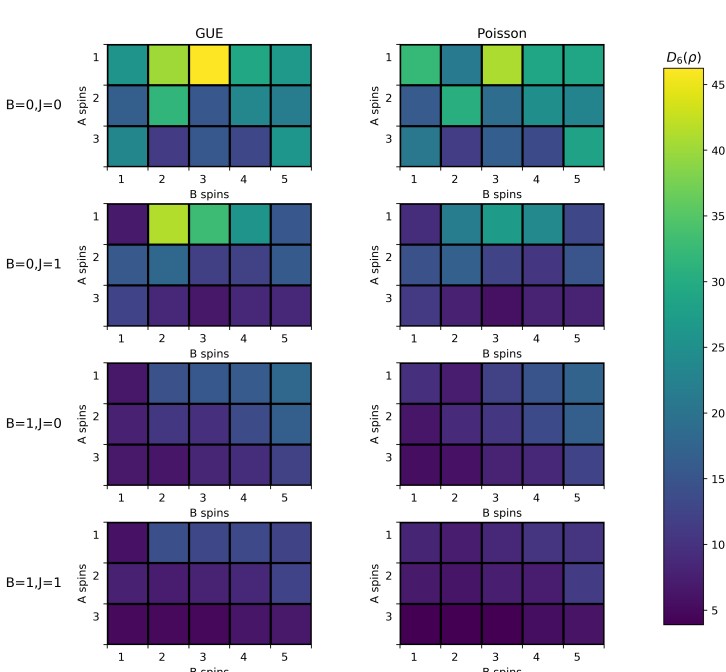

Figure 25: Left: Normalized distance of dynamics of XXZ Model (see equation 44) to analytical GUE. Right: to analytical Poisson.

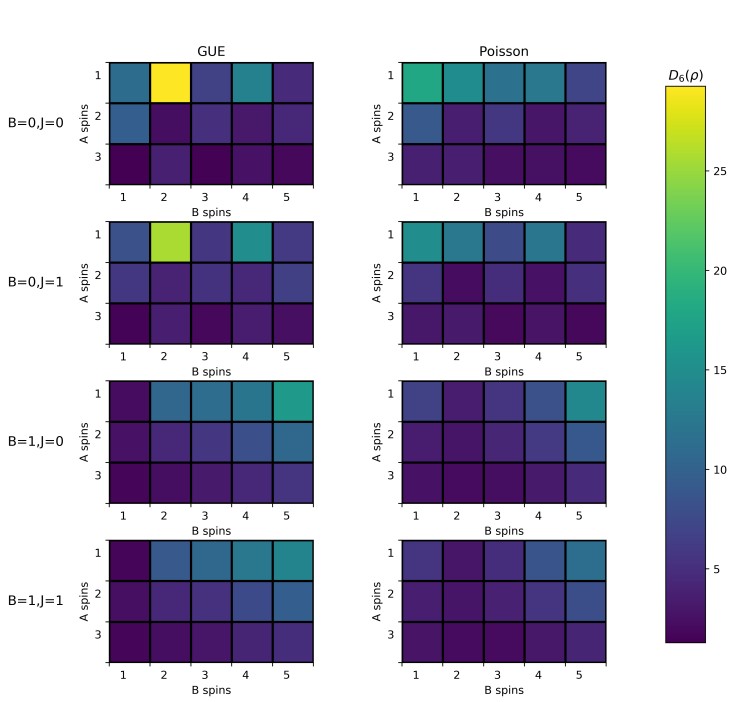

Figure 26: Left: Normalized distance of dynamics of Disordered XXZ Model (see equation 45) to analytical GUE. Right: to analytical Poisson.

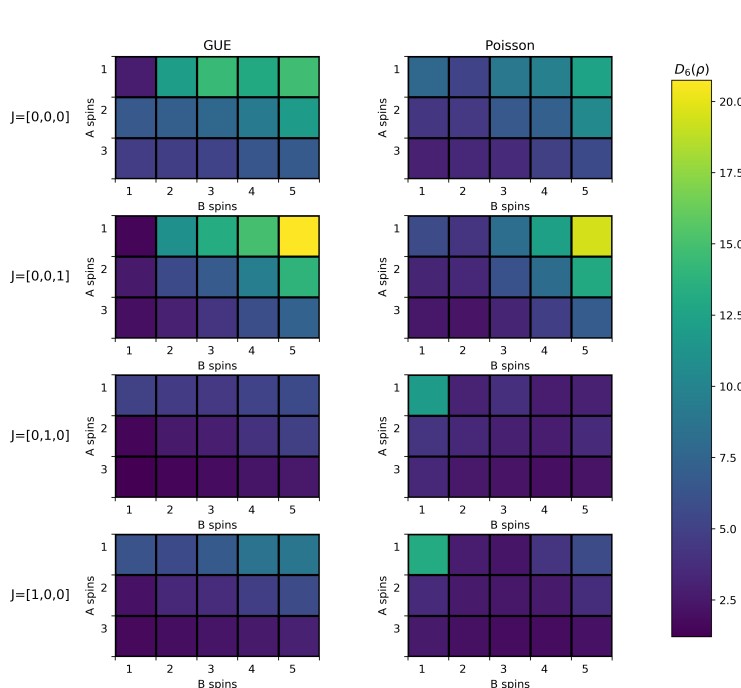

Figure 27: Left: Normalized distance of dynamics of Spin Glass Model (see equation 47) to analytical GUE. Right: to analytical Poisson.

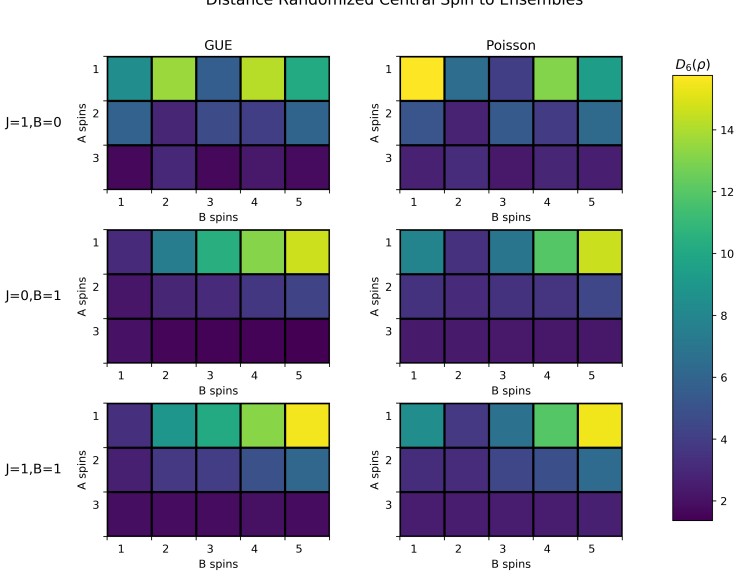

Figure 28: Left: Normalized distance of dynamics of Central Spin Model (see equation 48) to analytical GUE. Right: to analytical Poisson.

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
