# Peer review of "Entanglement Dynamics of Random GUE Hamiltonians"

_SciPost Physics Core, doi:SciPost Phys. 10, 071 (2021)_

## Round 2 · Referee Report · Anonymous (Referee 1) · 2020-2-18

Strengths
Weaknesses
Report
The paper by Chernowitz and Gritsev deals with the problem of computing the dynamics of the entanglement between two subsystems A,B under time evolution with an Hamiltonian H which is mimicked by a random Hermitian matrix, and therefore has no knowledge of space and in particular of the separation between A,B.
This setup is probably a good approximation for some observables of an ergodic system, but not for all, as locality has a strong bearing on several things. Also, it is probably good for dynamics sufficiently high in energy as going close to the ground state can be influenced by fine details of the hamiltonian. Anyway, as RMT proved many times to be a valuable tool for analysis of typical ergodic systems, it is definitely worth looking at this problem.
The authors compute the purity of the system, $\mathrm{Tr}(\rho_A^2(t))$ as an entanglement measure and give the expressions in terms of $d$, the dimension of the full space and $d_A,d_B$, the spaces of $A,B$ (hence $d=d_A d_B$). They give general formulae for the purity as a function of $d,t$ but explicit expressions only for some, small $d$. They also consider the scaling limit $d\to\infty$ with $\tau=t/\sqrt{d}$ fixed. The problem can be solved exactly as it reduces to the Fourier transform of the semicircle law (giving a Bessel function). They conjecture that only some terms in the various quantities survive in this limit, and I suspect that this is essentially the planar limit of the diagrams contributing to $\mathrm{Tr}(\rho^2)$. The emergence of planar diagrams in matrix models has been noticed in the 70's. For a recent work on this topic (with focus on entanglement dynamics) I would suggest the author to look at the paper "Universal entanglement of typical states in constrained systems," SC Morampudi, A Chandran, CR Laumann, Physical Review Letters 124 (5), 050602.
By the way, I must mention the authors do not refer much to the work linking RMT and entanglement. In particular, for example, I would expect that, in the limit $t\to\infty$, the statistic of purity would reduce to what has been calculated in a series of works including (but not limited to) P Facchi, et al. Physical review letters 101 (5), 50502 (2008) and Celine Nadal, Satya N. Majumdar & Massimo Vergassola, Journal of Statistical Physics, volume 142, pages 403–438 (2011). In some later paper the statistics of the entanglement entropy for a random state was calculated as well (the authors in the conclusion say they would like to use a Gross-Witten generating functional to this scope) and this should again be the limit of $t\to\infty$ of their results. I am not sure if they intended to do the calculation for finite $t$ with a GW functional.
The authors then do some numerics on some disordered models and (XXZ, TFIM, SYK) to compare with the analytic results of the previous sections. I must say that the disorder distribution used in Eq.s (42)-(44) is unusual (may have been used in other papers before but I have never seen it). For example, a typical choice of disorder for the XXZ model might take the form of $\sum_i J_{ij} \vec\sigma_i \cdot\vec\sigma_j$ with or without a term $\sum_i h_i\sigma^3_i$. What is the need for having the peculiar form with the $X_{j}^{2a}$? Maybe the authors have something in mind which makes this choice compelling. The numerics is performed but not discussed in much detail (what happens as the system size increases?) and it is not clear what the take home message of this section is.
All in all the paper is well written (not exceptionally well written, though) and the results are quite surely correct and (to my knowledge) original, but only marginally interesting. There is not much discussion of the previous works using RMT do describe the statistics of entanglement, but there is a large discussion of the literature dealing with integrable models, or ETH systems. I also do not understand the relevance of this work with the study of the stability of the MBL phase mentioned in the introduction. The author should argument better on this point.
I think, however, with some more work the paper should be publishable.
Requested changes
See weaknesses
Dear reviewer,
I (Daniel) will respond on both our behalves.
Thank you for the elaborate report. First off, I wish to apologize for leaving it unanswered for so long, while we focused on other projects.
Second, It is clear you have taken the time to really understand the story and the function of the various sections. For this I feel very grateful, as it means your criticism is well founded and useful. And mostly I agree that said criticism is fair. I especially appreciate your summary "All in all the paper is well written (not exceptionally well written, though) and the results are quite surely correct and (to my knowledge) original, but only marginally interesting." If I were honest, I probably could not summarize it better myself.
Thirdly, the report has been updated, and we have mainly focused, as you suggest, on fleshing out the section on the numerics of established models more. The manner of randomizing these ensembles is our own, so it is quite possible you have not seen it before. But you were correct to point out that the results were not completely carried to term. We have collected the comparisons of the various models into a single figure, which sheds some more quantitative light on which ones are well described by the GUE and which aren't. We also added the Central Spin, which was mentioned in the introduction but not used, and removed mentions of the MBL from the introduction as we didn't use it. This all made the piece more consistent.
As for the true 'thermodynamic limit' of the numerical models, we constructed these randomized ensembles ourselves and do not know how to push to infinite Hilbert space dimension with them in general, the XXZ, Ising, Spin Glass in particular. The implementations of the Central spin and SYK we took were conventional. These two are also very rich subjects with subtleties, and trying to treat the limits of subsystem dynamics for infinite dimensions of any one of them would be a reasonable project on its own. The point of this paper is not to be an encyclopedia of established models, but to compare, in the focus of small system sizes, our analytic results to a (frankly arbitrary) set of hopeful candidate models. In short: we can handle up to 8 qubits for all the models straightforwardly, but cannot access larger orders of magnitude numerically without greatly extending the scope of the work. We have predicted which models we expect to work well at large sizes, but it is merely conjecture.
Moving on to the remaining points of critique: I feel that there is indeed some discussion of the limits of our analytical results for large time or dimension. The low hanging fruit has at least been plucked. If it is important the reviewer, we can add more explicitly the limits of the various quotients as the system size increases, perhaps keeping the ratio of system to bath size fixed, but most readers with a grasp of calculus could do the same. The Bessel function scaling limit is explained, being the only tricky part.
Next, I agree that the various results on entanglement averages in RMT are very interesting. However, for instance the works of Facchi, or those of Majumdar et al, as I understand them, have to do with states that are drawn uniformly according to the Haar measure. We do compare our infinite time limit to these averages, and no, they are not exactly equal, however they are similar and approach each other in the infinite dimensional limit. I believe this comparison ultimately can only go so far: our work is dynamical and has a different setup: the evolving Hamiltonian is the random object, not the state.
And Finally, I think the notion that these calculations could be simplified by taking only the contributing planar diagrams is quite exciting. However, none of these calculations were done in a diagrammatic fashion, and to redo everything in another language in order to test this hypothesis, is a very ambitious project. One that might well allow for results in the thermodynamic limit, some of which are in fact already known. We have mentioned the possible connection to the planar limit, thank you for the suggestion, but I do not feel equipped to investigate this thoroughly at present time. That would be a whole new paper.
I hope you feel we have taken your criticism seriously, as we do. In so far as we have not complied, it is mainly because we don't see a clear path to and don't want to blow hot air that is not backed up with rigor. If you feel we should try again, we will.
Thank you again,

Author: Daniel Chernowitz on 2020-06-30 [id 866]
(in reply to Report 2 on 2020-02-24)Dear reviewer,
Thank you for the time you took to construct this report. I should apologize for taking so long to reply while I focused on other projects.
You are correct that the results for the unitary average are not new, we found out about [28] halfway along the project. And indeed, the late time averages of our expressions are readily predicted, nevertheless we calculated them. We have filled a small niche, of intermediate times in tiny systems, one that is of more mathematical than physical interest.
In the very beginning, the project was to analyze the entanglement (Von Neumann) entropy, as you suggest. However, those calculations quickly falter, because the matrix logarithm of the density matrix either requires diagonalization, or an infinite series, to calculate, and the RMT techniques we had at our disposal do not allow a closed for expression, or at least it is not tractable, to us. Therefore I agree that it would be a more interesting statistic, and look forward to the work of a more skilled researched who will average it. We have made due with the density matrix and purity.
Based on the reviews, the paper has been updated.
We have included more elaborate numerics, indeed up to 8 qubits, for both the main results, and for a number of established numerical models. These numerics help tie the analytic averages to more concrete parts of physics. We hope this is enough foundation to warrant a publication.
Thanks again,

---

## Round 2 · Referee Report · Anonymous (Referee 2) · 2020-2-24

Strengths
Weaknesses
2-Physical significance appears marginal
Report
In particular they study averages of the reduced density matrix and the purity over the group (U(d)) while the radial average over the eigenvalues is obtained exploiting known orthogonal polynomial techniques.
The analytical results for the group average are not new, they mainly extend Ref. [28].
On the other hand, the new analytical results for the eigenvalue averages for $n=1, 2$ do not have, in my opinion, a sufficient physical significance to warrant publication. The large-time behaviour of the reduced density matrix and the purity easily follow from the Riemann Lebesgue lemma and the regularity of the diagonal element of the kernel.
I suggest the authors to analyze numerically the behaviour of the entanglement entropy for an Hilbert space of a few qubits say $d=2^{8}$ $(d_A=d/2)$. Does it show sign of linear growth before saturating? And if not why? The more interesting case will be probably $d\rightarrow\infty$ with $d/d_A$ fixed but it might be too hard.
After this additional study I could recommend the paper in SciPost.
Requested changes
see report

---

## Round 3 · Referee Report · Anonymous (Referee 2) · 2021-3-3

Report

I am satisfied with the author's reply to my report. I recommend the paper for publication in Scipost.

---

## Round 3 · Referee Report · Anonymous (Referee 1) · 2021-3-11

Report

The authors have modified the paper in a satisfactory manner.

---

## Round 3 · List of Changes

We have included more elaborate numerics, up to 8 qubits, for both the main results, and for a number of established numerical models, and have made the comparisons to the GUE and Poissonian ensembles more tangible. These numerics help tie the analytic averages to more concrete parts of physics.

We have mentioned connections to planar limits of diagrams.

We have made the comparisons explicit to existing limits in RMT and entanglement of random quantum states.

---

## Editorial Decision

published